# Cardiomyocyte gene programs encoding morphological and functional signatures in cardiac hypertrophy and failure

Seitaro Nomura[1,2], Masahiro Satoh[2,3], Takanori Fujita[2], Tomoaki Higo[4], Tomokazu Sumida [1], Toshiyuki Ko[1], Toshihiro Yamaguchi[1], Takashige Tobita[5], Atsuhiko T. Naito[1], Masamichi Ito[1], Kanna Fujita[1], Mutsuo Harada[1], Haruhiro Toko[1], Yoshio Kobayashi[3], Kaoru Ito[6], Eiki Takimoto[1], Hiroshi Akazawa[1], Hiroyuki Morita[1], Hiroyuki Aburatani [2] & Issei Komuro[1]

Pressure overload induces a transition from cardiac hypertrophy to heart failure, but its underlying mechanisms remain elusive. Here we reconstruct a trajectory of cardiomyocyte remodeling and clarify distinct cardiomyocyte gene programs encoding morphological and functional signatures in cardiac hypertrophy and failure, by integrating single-cardiomyocyte transcriptome with cell morphology, epigenomic state and heart function. During early hypertrophy, cardiomyocytes activate mitochondrial translation/metabolism genes, whose expression is correlated with cell size and linked to ERK1/2 and NRF1/2 transcriptional networks. Persistent overload leads to a bifurcation into adaptive and failing cardiomyocytes, and p53 signaling is specifically activated in late hypertrophy. Cardiomyocyte-specific p53 deletion shows that cardiomyocyte remodeling is initiated by p53-independent mitochondrial activation and morphological hypertrophy, followed by p53-dependent mitochondrial inhibition, morphological elongation, and heart failure gene program activation. Human single-cardiomyocyte analysis validates the conservation of the pathogenic transcriptional signatures. Collectively, cardiomyocyte identity is encoded in transcriptional programs that orchestrate morphological and functional phenotypes.

[1] Department of Cardiovascular Medicine, Graduate School of Medicine, The University of Tokyo, Tokyo 113-8655, Japan. [2] Genome Science Division, Research Center for Advanced Science and Technologies, The University of Tokyo, Tokyo 153-0041, Japan. [3] Department of Cardiovascular Medicine, Chiba University Graduate School of Medicine, Chiba 260-8670, Japan. [4] Department of Cardiovascular Medicine, Osaka University Graduate School of Medicine, Osaka 565-0871, Japan. [5] Department of Cardiology, Tokyo Women's Medical University, Tokyo 162-8666, Japan. [6] Laboratory for Cardiovascular Diseases, RIKEN Center for Integrative Medical Sciences, Kanagawa 230-0045, Japan. These authors contributed equally: Seitaro Nomura, Masahiro Satoh. Correspondence and requests for materials should be addressed to H.A. (email: haburata-tky@umin.ac.jp) or to I.K. (email: komuro-tky@umin.ac.jp)

Organs respond appropriately to external and internal stress to maintain homeostasis, but excessive stress disrupts the adaptive response, leading to organ dysfunction. Hemodynamic stimuli such as pressure and volume overload to the heart initially induce cardiac hypertrophy as an adaptive response to reduce wall stress and prevent cardiac dysfunction[1,2]. However, sustained overload causes cardiac dysfunction leading to heart failure[3–5]. During this process, cardiomyocytes activate various signaling cascades initially for adaptive morphological hypertrophy, followed by a transition to the failing phenotype characterized by elongation and contractile force reduction[6]. Yet, it remains elusive how individual cardiomyocytes undergo molecular and morphological remodeling in response to stress, contributing to cardiac adaptation and dysfunction.

Because individual cardiomyocytes constitute the basic units of gene regulation, each cardiomyocyte's phenotype and function are considered to be determined based on its transcriptional programs. Chemical inhibition of transcriptional activation has been shown to suppress cardiac molecular and morphological remodeling after pressure overload[7]. Single-cardiomyocyte gene expression analyses have revealed an increase in cell-to-cell transcriptional variation in the aging mouse heart[8] and partial activation of genes involved in de-differentiation and the cell cycle[9]. These studies have established that cardiomyocyte gene expression underlies cellular phenotypes and determines cardiac function, but it remains elusive what gene programs regulate morphological remodeling and contribute to maintain and disrupt cardiac homeostasis. Furthermore, to reveal the pathogenesis of heart failure and identify novel therapeutic targets, it is important to better understand the molecular and morphological bases of the hypertrophic and failing cardiomyocyte states and to identify regulators of the transition between these states. Uncovering the conserved gene programs involved in cardiomyocyte morphology and cardiac function will enable accurate assessment of the condition of cardiomyocytes and the heart and prediction of treatment response.

Here, through co-expression network analysis[10,11] of mouse and human single-cardiomyocyte transcriptomes, we integrated gene expression profiles with multidimensional data such as single-cell morphology, epigenomic state, and physiological heart function to dissect the molecular and morphological dynamics of cardiomyocytes during cardiac hypertrophy and heart failure. Our study establishes that cardiomyocyte identity is encoded in transcriptional programs that orchestrate morphological and functional phenotypes, and can be controlled by appropriate interventions.

## Results

**Single-cardiomyocyte transcriptomic profiles in heart failure.**
We obtained single-cardiomyocyte transcriptomes from mice exposed to pressure overload (Fig. 1a). In our model, transverse aorta constriction (TAC)[12,13] in 8-week-old C57BL/6 mice induced cardiac hypertrophy at 1–2 weeks after the operation and heart failure at 4–8 weeks (Fig. 1a and Supplementary Fig. 1a, b). Pressure overload increased the cellular width-to-length ratio in early cardiac hypertrophy, which is a morphological feature of cardiomyocytes of the concentric hypertrophic heart[14] (Fig. 1b). After establishing the single-cardiomyocyte transcriptome analysis pipeline (Supplementary Fig. 1c, d), we isolated cardiomyocytes from the left ventricular free wall after sham operation and at 3 days and 1, 2, 4, and 8 weeks after TAC using Langendorff perfusion (Fig. 1a) and prepared 540 single-cardiomyocyte cDNA libraries with the SMART-seq2 protocol[15]. After removing 58 libraries showing incomplete reverse transcription (Supplementary Fig. 1e–g), we sequenced

482 libraries and obtained 396 single-cardiomyocyte transcriptomes (sham, 64 cardiomyocytes; day 3, 58; week 1, 82; week 2, 61; week 4, 73; and week 8, 58) in which >5000 genes were detected (RPKM > 0.1; Supplementary Fig. 1h–n and Supplementary Data 1 and 2). Averaged single-cell expression profiles were tightly correlated with the corresponding bulk expression profiles (Supplementary Fig. 1o, p). Averaged single-cell profiles were also highly correlated between biological replicates (Supplementary Fig. 1q, r), whereas expression profiles substantially differed between individual cells even in the same heart (Supplementary Fig. 1s).

Hierarchical clustering revealed that cardiomyocytes from sham and TAC mice were well clustered (Supplementary Fig. 2a). Principal component analysis showed that cardiomyocytes from different batches at the same time points were located close to each other (Supplementary Fig. 2b). Principal component 1 (PC1) values were correlated with the number of expressed genes and cells with high PC1 values were enriched for cardiomyocytes from mice at 3 days and 1 week after TAC operation (Supplementary Fig. 2c). Differential expression analysis revealed that genes encoding proteins that function in the mitochondria, ribosomes, endoplasmic reticulum, and cytoskeleton were highly expressed at 3 days and 1 week after TAC operation (Supplementary Fig. 3a). We consider that the increase of the number of detected genes at the early stage after pressure overload might reflect the expression of these stress response genes and/or an enlargement of cell size after pressure overload.

Hierarchical clustering also revealed that some genes thought to be specifically expressed in endothelial cells (e.g., Cav1 and Pecam1) or fibroblasts (e.g., Dcn and Lum) were clustered together with genes essential for transcription; Cav1 and Pecam1 were in G7 and Dcn and Lum were in G8 (Supplementary Fig. 2a). We used single-molecule RNA in situ hybridization (smFISH)[16] to confirm that these genes are expressed in sham and TAC cardiomyocytes (Supplementary Fig. 4). A previous study of single-nucleus RNA-seq of cardiomyocytes also mentioned the presence of cardiomyocytes expressing endothelial marker genes[9], consistent with our findings. Therefore, we did not eliminate cardiomyocytes expressing these genes from the downstream analysis.

Cell-to-cell transcriptional heterogeneity increased after pressure overload (Fig. 1c). The expression levels of certain genes in the heart failure stage varied considerably (e.g., Nppa, Myh7, and Xirp2; Supplementary Fig. 5a, b). We used smFISH to quantify single-cell mRNA levels of Myh7 (high variability) and Atp2a2 (low variability), confirming gene expression distributions similar to those obtained from single-cardiomyocyte RNA-seq analysis (Supplementary Fig. 5c–e and Supplementary Data 3).

**Network analysis of single-cardiomyocyte transcriptomes.** We applied weighted gene co-expression network analysis (WGCNA)[10] to all single-cardiomyocyte transcriptomes to identify 55 co-expression gene modules (Supplementary Fig. 6a and Supplementary Data 4) and calculated module eigengene expression profiles, which are the first principal components of a given module, across all cardiomyocytes. Unsupervised hierarchical clustering of all modules divided cardiomyocytes into 10 clusters (Supplementary Fig. 6b), but classification accuracy was not high (Supplementary Fig. 6c). To construct a better model, we used the Random Forests machine learning algorithm[17] to identify nine modules significant for cell classification (Supplementary Fig. 6d). These modules were active in many cardiomyocytes, whereas the other modules were active in a few cardiomyocytes (Supplementary Fig. 6b). Through the combination of hierarchical clustering and Random Forests analyses, we classified

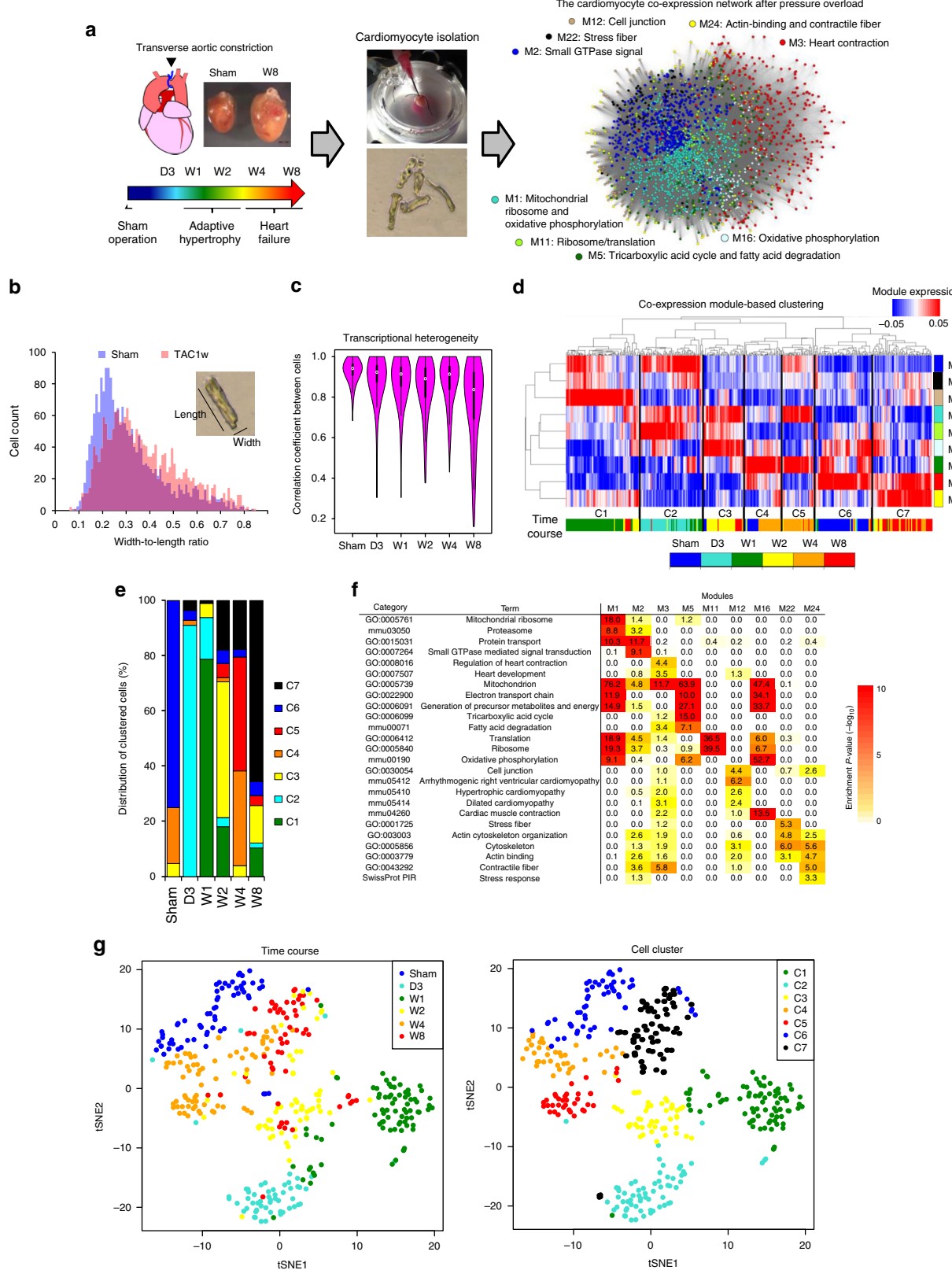

cardiomyocytes into seven clusters (Fig. 1d), resulting in highly accurate classification (Supplementary Figs. 6c and 7). We calculated the proportions of clustered cells to reveal cell-type heterogeneity at each time point (Fig. 1e). Gene ontology (GO) analysis showed that each module was enriched for genes involved in specific functions, such as translation, metabolism, heart contraction, and cell junction (Fig. 1f). Therefore, we named each module according to its most characteristic terms (Fig. 1a).

Co-expression gene network analysis revealed a structure highly clustered with these modules (Fig. 1a). Among the nine

**Fig. 1** Co-expression network analysis of single-cardiomyocyte transcriptomes. **a** Experimental scheme to construct the co-expression network of single-cardiomyocyte transcriptomes from mice exposed to pressure overload. D3 day 3, W1 week 1, W2 week 2, W4 week 4, W8 week 8. In the network, nodes indicate genes and are positioned according to the weighted prefuse force-directed layout algorithm in Cytoscape. Edges indicate a significant correlation between genes. Length of the edges is relative to the expression similarity of the connected genes, with a short edge corresponding to a high co-expression value. Dot colors indicate module colors, matching the colors in **d**. **b** Bar plot showing the distribution of the width-to-length ratio of cells from mice after sham and transverse aorta constriction (TAC) operation (W1). **c** Violin plot showing the distribution of the correlation coefficients of single-cell transcriptomes among cells at each time point. **d** Unsupervised co-expression module-based clustering classifying all cardiomyocytes (n = 396) into seven cell clusters (C1–C7). The colored bar below the heatmap indicates the time when the cardiomyocytes were extracted; the colored bar on the right indicates module colors matching the node colors of the network in **a**. **e** Bar plot showing the distribution of clustered cells at each time point. **f** Heatmap showing the enrichment of gene ontology (GO) and Kyoto Encyclopedia of Genes and Genomes (KEGG) pathway terms in each module. **g** t-Distributed stochastic neighbor embedding (t-SNE) visualization of cardiomyocytes clustered in **d**. Cells (dots) are colored by the time when cardiomyocytes were extracted (left) and according to the cell clusters in **d** (right)

modules, M1, M2, and M3 were especially well clustered. We conducted subnetwork analysis to identify the hub genes in each module: *Mrpl42*, *Cfl2*, and *Timm17a* for M1 (mitochondrial ribosome and oxidative phosphorylation), *Pdlim5* and *Eif5a* for M2 (small GTPase signal), and *Atp2a2* for M3 (heart contraction) (Supplementary Fig. 8a). Expression of these hub genes was critically correlated with that of the corresponding modules (Supplementary Fig. 8b), indicating that hub gene expression reflects the corresponding module activity.

We used t-distributed stochastic neighbor embedding (t-SNE) dimensionality reduction[18] to visualize the relationships among cells (Fig. 1g and Supplementary Fig. 9a). Cardiomyocytes assigned to the same clusters and those isolated at the same time points were located close to each other. Their transcriptomic profiles changed drastically after TAC operation, and returned to the baseline state slowly thereafter, but represented a state distinct from that of sham cardiomyocytes in the heart failure stage. By plotting module expression on the t-SNE map, we found modules expressed in specific cell clusters (Supplementary Fig. 9b). Consistent with this result, differential expression analysis revealed that *Atp2a2*, an M3 hub gene, was highly expressed in C6 and C7, whereas *Mrpl42*, an M1 hub gene, was not expressed specifically in C6 or C7 (Supplementary Fig. 3b). Module-to-module correlation analysis showed a negative correlation between M1 and M3 expressions (Supplementary Fig. 10a, b), These results suggest that the balanced expression of mitochondrial translation/metabolism genes and heart contraction genes is an innate feature of cardiomyocytes. We also found M24 specific expression in C7 cells (Fig. 1g and Supplementary Fig. 9b), most of which were cardiomyocytes at the heart failure stage (Fig. 1e), suggesting that M24 is characteristic of failing cardiomyocytes.

**Trajectory of cardiomyocyte remodeling**. We next used Monocle[19,20] to reconstruct the branched trajectory of cardiomyocyte remodeling after pressure overload, which consisted of three cell states and one branch point (Fig. 2a). Cardiomyocytes of the early stage after TAC operation (D3 and W1) mainly belonged to State 1 (Fig. 2b), suggesting that cardiomyocytes enter into State 1 immediately after TAC operation and were bifurcated into distinct cell fates (States 2 and 3) after the branch point. This bifurcation is considered to be related to the increased cell-to-cell heterogeneity (Fig. 1c). Almost all C7 cells were contained in State 3, suggesting that bifurcation into State 3 corresponds to the induction of failing cardiomyocytes.

Pseudo-time analysis revealed the transcriptional dynamics during the trajectory of cardiomyocyte remodeling (Fig. 2c, d). M2 (small GTPase signaling) and M22 (stress fiber formation) were specifically activated during the short period immediately after pressure overload. Module-to-module correlation analysis validated a strong correlation between M2 and M22 (Supplementary Fig. 10a, b). In addition, M1, M11, and M12 were

activated, whereas M3 and M5 were repressed after pressure overload (Fig. 2c). The strong correlation between M1 (mitochondrial ribosome and oxidative phosphorylation) and M11 (ribosome and translation) (Supplementary Fig. 10a, b) suggests the synchronized activation of nuclear and mitochondrial translation after pressure overload.

Cardiomyocytes near the branch point were mainly from mice at 2 or more weeks after TAC operation (Fig. 2a), suggesting that the bifurcation occurs in the late stage of hypertrophy. Module expression dynamics along the pseudo-time line (Fig. 2c) showed that the induction of State 2 (cardiomyocytes that escaped from the induction into failing cardiomyocytes) is characteristic for the activation of M5 (tricarboxylic acid cycle and fatty acid degradation). Conversely, the induction of State 3 (failing cardiomyocytes) is characteristic for the activation of M3 (heart contraction) and M24 (actin-binding and contractile fiber) and repression of M1 (mitochondrial ribosome and oxidative phosphorylation). Differential expression analysis with pseudo-time analysis confirmed that M3 genes (e.g., *Atp2a2*, *Myl2*, and *Myl3*) and M24 genes (e.g., *Ttn*, *Myom1*, and *Alpk3*) were activated, whereas M1 genes (e.g., *Mrpl42*, *Cfl2*, and *Timm17a*) were repressed in the induction of State 3 (Fig. 2d). M1 activation and M3 repression, which were observed in State 1, were abolished, suggesting the disruption of adaptive hypertrophy responses at the induction of failing cardiomyocytes. We used smFISH with an mRNA probe for *Atp2a2* (M3 hub gene) to validate the temporal dynamics of M3 activity (Supplementary Fig. 10c and Supplementary Data 5). M24 (actin-binding and contractile fiber) was activated specifically in State 3, suggesting the transcriptional signature of failing cardiomyocytes.

**Modules and regulators of cardiomyocyte hypertrophy**. We next linked the morphological and molecular heterogeneities in cardiomyocyte hypertrophy. To identify the gene modules involved in cardiomyocyte hypertrophy, we again isolated cardiomyocytes from mice at 1 week after TAC (early stage of cardiac hypertrophy), measured their area, obtained their transcriptomes, and integrated them (Fig. 3a and Supplementary Fig. 11a–e). Principal component analysis of their transcriptomes placed cardiomyocytes (Fig. 3b) and revealed that PC1 was positively correlated with cell area and M1 (mitochondrial ribosome and oxidative phosphorylation) and negatively correlated with M3 (heart contraction) (Fig. 3b–e). Correlation analysis between module expression and cell area showed that M1 and M5 (tricarboxylic acid cycle and fatty acid degradation) were strongly correlated with cardiomyocyte area (Fig. 3f). The expression of M1 hub genes (*Mrpl42*, *Timm17a*, and *Cfl2*; Fig. 3g) was also positively correlated with cardiomyocyte area (Fig. 3h and Supplementary Data 6). GO analysis showed the significant enrichment of mitochondrial genes among the genes correlated with cell area (Fig. 3i and Supplementary Fig. 11f). These results indicate

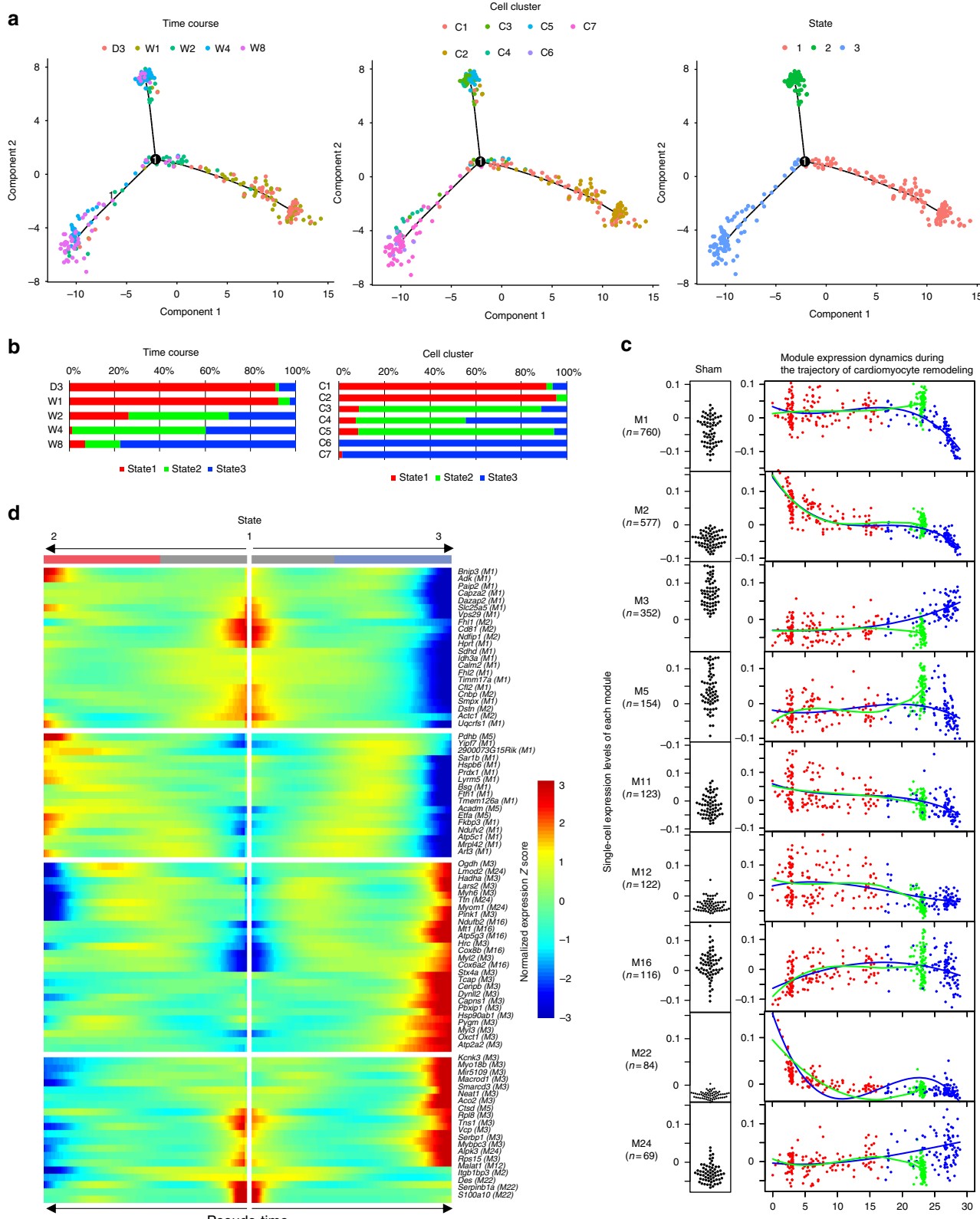

**Fig. 2** Single-cardiomyocyte trajectory analysis. **a** Single-cardiomyocyte trajectory after TAC operation reconstructed by Monocle. Cells are colored by the time when cardiomyocytes were extracted (left), according to the cell clusters (middle), and by the cell states annotated by Monocle (right). **b** Bar plot showing the distribution of the cell states (State 1, State 2, and State 3) at each time point (left) and in each cell cluster (right). **c** Module expression dynamics along the pseudo-time reconstructed by Monocle. Dot colors indicate state colors, matching the colors in **b**. Single-cell expression levels of each module in sham cardiomyocytes are represented as black dots (left). The fitted curves for the trajectory into State 2 (green) and State 3 (blue), and the number of genes assigned to the corresponding modules are also shown. **d** Heatmap showing the expression levels of differentially expressed genes during the trajectory. Corresponding modules are also denoted in parentheses

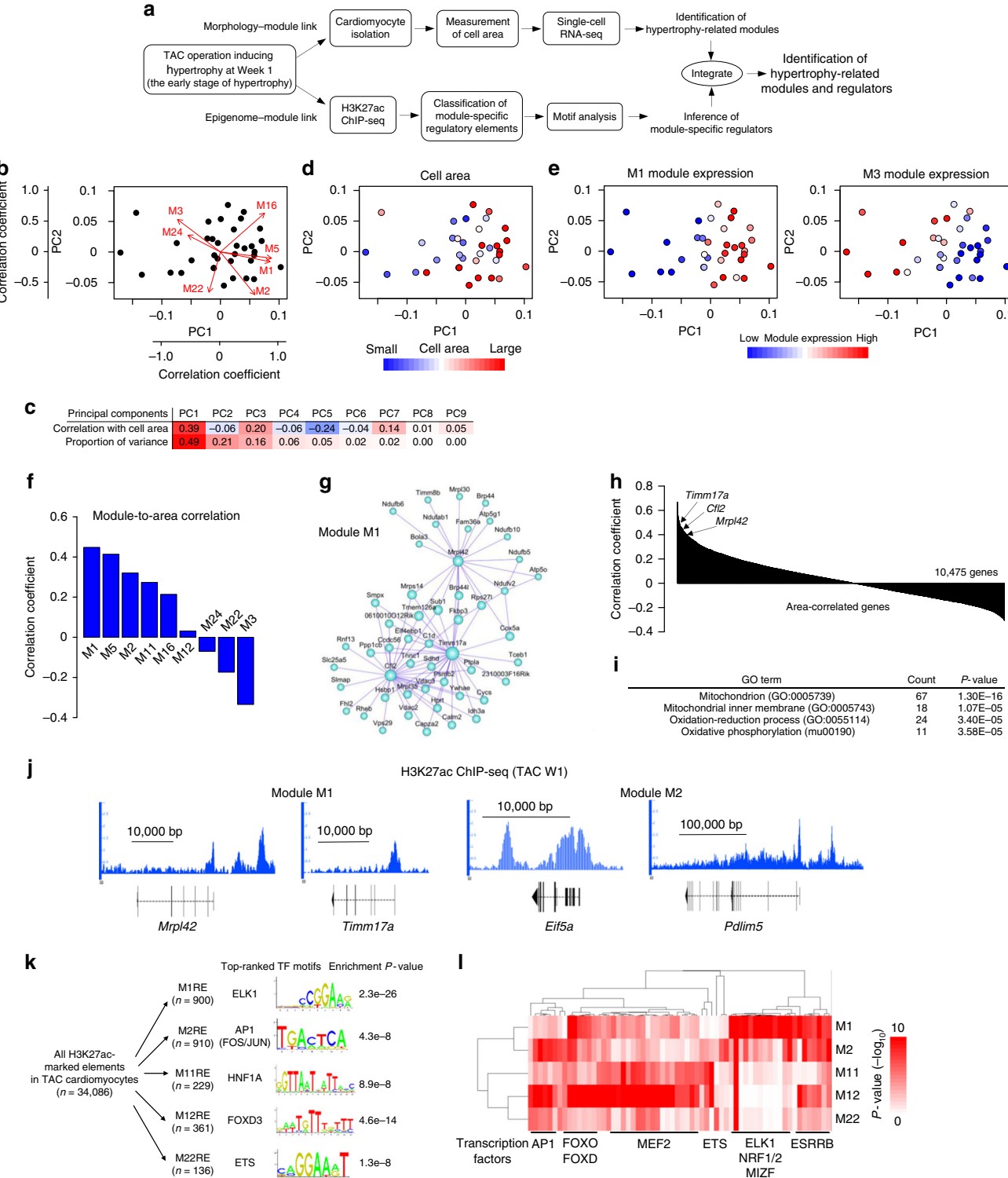

**Fig. 3** Hypertrophy-related modules and regulatory factors. **a** Experimental scheme to identify hypertrophy-related modules and regulators. **b** Principal component analysis (PCA) plot of cardiomyocytes from mice at 1 week after TAC in **a**. Arrows denote the correlation coefficients of the respective module with each principal component. **c** Correlation coefficient with cell area and proportion of variance for each principal component. **d, e** PCA plots colored by the cell area (**d**) and by the expression of each module (**e**). **f** Bar plot showing the correlation coefficient between cell area and module expression. Modules are ordered by the correlation levels. **g** Hub gene network of M1. The size of the dots represents node centrality. **h** Bar plot showing the correlation coefficient between cell area and gene expression. Genes whose expression was detected in at least one of the samples are ordered by correlation levels. **i** List of the most enriched GO terms in the top 300 correlated genes with cell area. **j** Representative genome browser views of H3K27ac ChIP-seq of cardiomyocytes from mice at 1 week after TAC. The *Y*-axis indicates reads per million (range, 0–3). **k** Transcription factor recognition motifs most significantly enriched in the regulatory elements (REs) for each module. **l** Hierarchical clustering of transcription factor recognition motifs. The top 10 significantly enriched motifs for each module are selected. Transcription factors strongly expressed in cardiomyocytes are shown below the heatmap

that the expression of mitochondrial ribosome and oxidative phosphorylation genes is closely linked to the extent of cell hypertrophy. Mitochondrial biogenesis increases during cardiac hypertrophy[21] and cell size correlates with the number of mitochondria[22,23]. The transcription and translation of nuclear genes involved in mitochondrial oxidative phosphorylation are induced rapidly in response to external stimuli during mitochondrial biogenesis[24], suggesting that pressure overload-induced M1 activation might participate in mitochondrial biogenesis to match the increased energy demand during cardiac hypertrophy.

To uncover the regulatory mechanisms regulating cardiomyocyte hypertrophy, we performed chromatin immunoprecipitation followed by sequencing (ChIP-seq) for histone H3 acetylated lysine 27 (H3K27ac) modification[25] in cardiomyocytes from mice at 1 week after TAC (Fig. 3a, j). We divided H3K27ac-marked elements into putative regulatory elements for each module and evaluated the enrichment of transcription factor recognition motifs to infer the regulatory factors for each module (Fig. 3k). Furthermore, the clustering of motif enrichment for the top 10 enriched transcription factors of each module revealed specific combination patterns of transcription factors and their regulating modules (Fig. 3l and Supplementary Fig. 12). Recognition motifs of ETS domain-containing protein (ELK1) and nuclear respiratory factors 1 and 2 (NRF1/2) were specifically enriched around M1 genes. Extracellular signal-regulated protein kinases 1 and 2 (ERK1/2) signaling induces cardiomyocyte hypertrophy[26] via ELK1 phosphorylation[27] and NRF1/2 signaling enhances mitochondrial biogenesis[28,29]. Collectively, the combination of single-cardiomyocyte RNA-seq and H3K27ac ChIP-seq suggests that the ERK1/2 and NRF1/2 signaling pathways coordinately regulate the mitochondrial ribosome and oxidative phosphorylation module, which is critical for cardiomyocyte hypertrophy.

**Signaling pathway inducing heart failure**. We investigated how cardiomyocytes in the adaptive hypertrophy stage become failing cardiomyocytes. The characteristic module dynamics, such as M1 repression and M3 activation, were observed from the later stage of adaptive hypertrophy (Fig. 2). Heart failure stage-specific cells (C7 cells) began to appear from this stage (Fig. 1e). Thus, we speculated that the activation of some signaling pathway(s) in hypertrophy-stage cardiomyocytes might participate in the transition from hypertrophy to failing cardiomyocytes and the development of heart failure. Therefore, we sought to identify the hypertrophy stage-specific modules activated in even a small population of cardiomyocytes among all modules. If there are hypertrophy stage-specific networks, network analysis using single-cell transcriptomes of all cardiomyocytes except hypertrophy-stage cardiomyocytes cannot identify these network modules. This idea is based on the concept for identifying sample-specific networks[30,31]. Therefore, we separately performed WGCNA on all cardiomyocytes and on those except hypertrophy-stage cardiomyocytes, and assessed the significance of the overlap between gene modules. We identified 12 hypertrophy stage-specific modules (Fig. 4a and Supplementary Fig. 13a), all of which were active in only a few cardiomyocytes (Supplementary Fig. 6b). Pathway enrichment analysis of these 12 modules revealed that genes involved in the cell cycle and p53 signaling pathways were enriched in M7 (Fig. 4b). smFISH showed that mRNA of Cdkn1a, an M7 network component (Fig. 4c), was specifically highly expressed in certain hypertrophy-stage cardiomyocytes (Fig. 4d, e and Supplementary Data 7). GO analysis revealed that M7 was also enriched for genes involved in the DNA damage response (Fig. 4f). Immunostaining of p21 (encoded by Cdkn1a) and phospho-histone H2A.X (DNA damage response marker; γH2A.X)[32] demonstrated that p21 was

specifically detected in hypertrophy-stage cardiomyocytes and that p21-positive cardiomyocytes were also γH2A.X-positive (Fig. 4g and Supplementary Fig. 13b, c). Thus, our unbiased single-cell network analysis uncovered DNA damage-induced p53 signaling activation in hypertrophy-stage cardiomyocytes.

Since the accumulation of DNA damage induced by oxidative stress promotes heart failure progression[33–35], we hypothesized that the oxidative stress response might be activated at the hypertrophy stage and be associated with morphological hypertrophy. GO analysis showed that M1 and M5 were specifically enriched for genes involved in the oxidative stress response (Supplementary Fig. 13d). Among these genes, peroxiredoxins, oxidative stress biomarkers[36], were critically correlated with cell area during the early stage of hypertrophy (Supplementary Fig. 13e). These results suggest that morphological hypertrophy after pressure overload is involved in both enhanced mitochondrial biogenesis for increased energy demand and excessive oxidative stress leading to the accumulation of DNA damage.

DNA damage accumulation, observed in patients with heart failure[35], has been reported as a cause of heart failure progression[34]. Although p53 has been reported to be detrimental for the heart[12,37], its role in cardiomyocyte remodeling during heart failure remains elusive.

**p53-induced molecular and morphological remodeling**. To assess whether p53 activation in cardiomyocytes is critical for the transition to failing cardiomyocytes and development of heart failure, we generated cardiomyocyte-specific p53 knockout (p53CKO) mice (Supplementary Fig. 13f). Baseline features did not differ significantly between p53$^{flox/flox}$ and p53CKO mice (Fig. 5a, b and Supplementary Data 8). Echocardiography demonstrated that p53$^{flox/flox}$ and p53CKO mice showed similar cardiac hypertrophy at 2 weeks after pressure overload (Fig. 5a, b and Supplementary Data 8). However, at 4 weeks after pressure overload, cardiac function was impaired in p53$^{flox/flox}$, but not in p53CKO mice (Fig. 5a, b and Supplementary Data 8).

We assessed the detailed molecular mechanism of p53 in cardiomyocytes by single-cardiomyocyte transcriptome analysis of p53$^{flox/flox}$ and p53CKO mice at 2 weeks after pressure overload, when the transition from hypertrophic to failing cardiomyocytes occurs (Figs. 1e and 2a). Module expression analysis revealed that p53 downregulated the mitochondrial ribosome and oxidative phosphorylation (M1 and M16) and small GTPase signaling (M2) modules, and upregulated the heart contraction (M3), cell junction (M12) and actin-binding and contractile fiber (M24) modules (Fig. 5c). We used hierarchical clustering and t-SNE to classify and organize these cardiomyocytes into transcriptionally distinct clusters (Fig. 5d, e), and identified differentially expressed modules (Fig. 5f). M1-low, M3-high, and M24-high cardiomyocytes (cluster A; Fig. 5d), which correspond to failing cardiomyocytes (C7 cells; Figs. 1d, e and 2c, d), appeared in p53$^{flox/flox}$ mice, but rarely in p53CKO mice (Fig. 5d–g). Allocation of cardiomyocytes after pressure overload to predetermined cell clusters using the machine learning Random Forests algorithm[17] indicated that cardiomyocyte remodeling progressed to the C5 and C7 stages in p53$^{flox/flox}$ mice but stopped at the C3 stage in p53CKO mice (Figs. 1d, e and 5h; Fisher's exact test, $P < 0.05$), suggesting that p53 activation is necessary for the induction of failing cardiomyocytes (C7 cells). Increased cell-to-cell transcriptional heterogeneity after pressure overload was diminished by p53 disruption (Fig. 5i). These results indicate that p53 in hypertrophy-stage cardiomyocytes is essential for molecular remodeling into failing cardiomyocytes by disrupting the

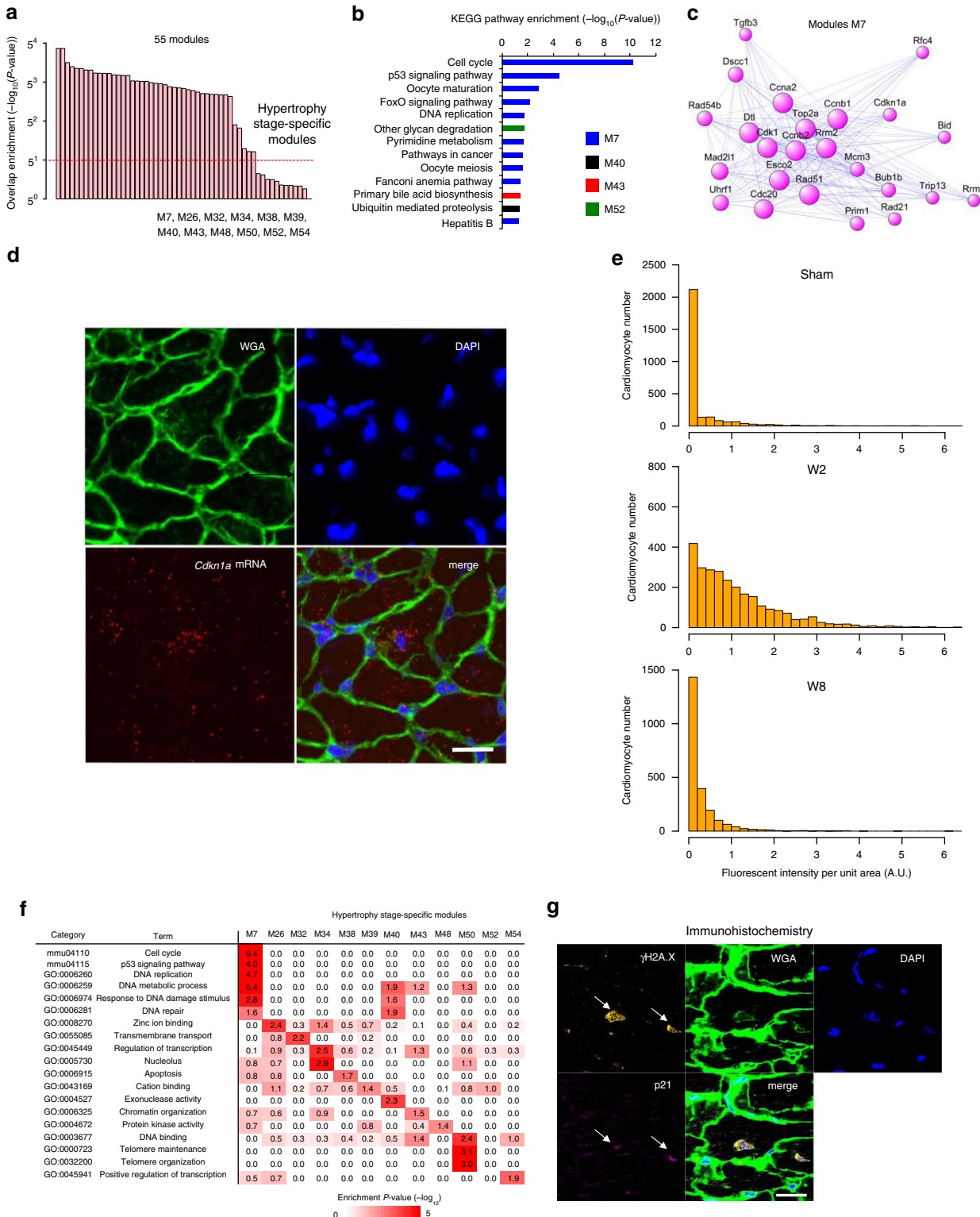

**Fig. 4** Hypertrophy-stage-specific p53 signaling activation. **a** Bar plot showing the statistical significance of the overlap between modules detected from co-expression analysis with or without hypertrophy-stage cardiomyocytes. The most strongly overlapping module pairs are selected from Supplementary Fig. 13a. Modules are ordered by significance level. The red line indicates the threshold for hypertrophy stage-specific modules. **b** KEGG pathway enrichment of 12 hypertrophy stage-specific modules identified in **a** and Supplementary Fig. 13a. **c** Co-expression network analysis of M7. **d** Representative images of *Cdkn1a* mRNA smFISH in the heart from mice at 2 weeks after pressure overload. Wheat germ agglutinin (WGA) and DAPI are used as markers of the plasma membrane and nucleus, respectively. Scale bar, 20 μm. **e** Bar plots showing the distribution of cells corresponding to the single-cell fluorescent intensity of *Cdkn1a* mRNA detected by smFISH in the heart after sham and TAC (weeks 2 and 8) operation. **f** Heatmap showing the enrichment of GO and KEGG pathway terms in 12 hypertrophy stage-specific modules. **g** Immunohistochemical staining of gH2A.X and p21 in the heart of mice at 2 weeks after pressure overload. WGA and DAPI are used as markers of the plasma membrane and nucleus, respectively. Arrows indicate the nuclei of gH2A.X and p21 double-positive cardiomyocytes. Scale bar, 20 μm

adaptive hypertrophy modules and activating the heart failure modules.

M24 is characteristic for failing cardiomyocytes, which are induced by p53 signaling activation. Co-expression network analysis identified *Ttn*, *Xirp2*, and *Kif5b* as central genes of M24

(Fig. 6a). Single-cardiomyocyte RNA-seq validated the heterogeneous downregulation of these genes in p53CKO cardiomyocytes (Fig. 6b). Epigenomic profiling of murine cardiomyocytes in the heart failure stage revealed the enrichment of binding motifs of myocyte enhancer factor-2 (MEF2) and nuclear factor-

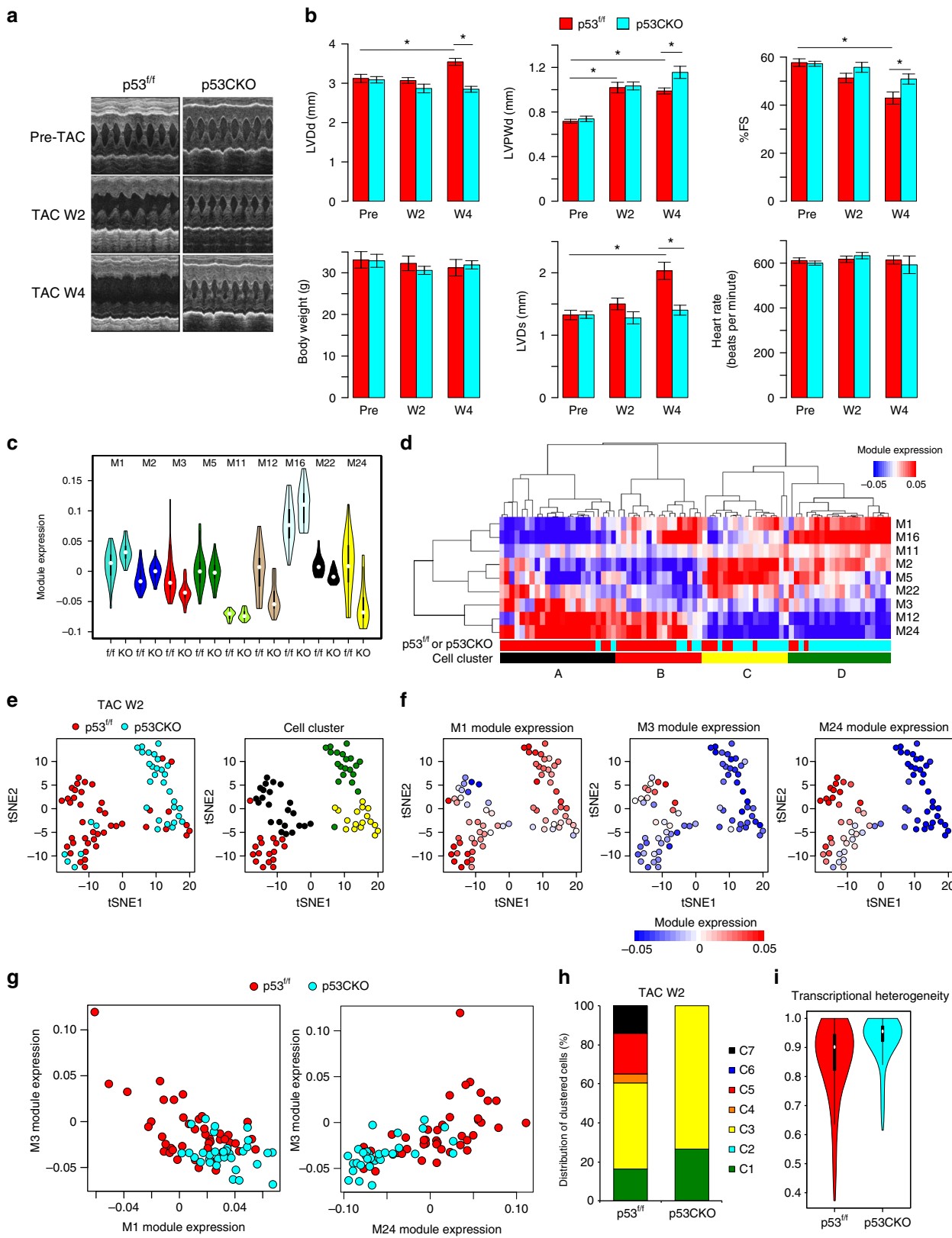

erythroid 2-related factor-2 (Nrf2) at active regulatory elements around M24 genes (Fig. 6c, d). MEF2 has been reported to interact with myocardin-related transcription factor to regulate actin dynamics[38] and mediate pathological remodeling into heart failure[39,40]. Nrf2, which regulates an antioxidant stress response[41,42], mediates maladaptive cardiac remodeling[43]. Consistent with this, GO analysis showed that M24 was specifically enriched for genes involved in both cytoskeleton organization (e.g., *Dst*, *Xirp2*, and *Ttn*) and stress responses (e.g., *Hsp90aa1*, *Hspa1a*, and *Hspa1b*) (Fig. 1f). These results suggest that failing cardiomyocytes induced by p53 activation are characteristic for MEF2-mediated cytoskeletal reorganization and Nrf2-mediated antioxidant stress responses.

We performed western blot analysis of heart tissues from p53[flox/flox] and p53CKO mice after sham operation and at 8 weeks after TAC operation (heart failure stage). We confirmed that Nrf2 protein expression was increased after TAC operation in p53[flox/flox] mice, which was blocked in p53CKO mice (Fig. 6e, Supplementary Fig. 14a–d, and Supplementary Data 8). This finding suggests that p53 mediates Nrf2 protein activation during heart failure. Since public Nrf2 ChIP-seq data of macrophages showed that Nrf2 binds to the Mef2a promoter region, which contains an Nrf2 recognition motif[44] (Supplementary Fig. 14e), we hypothesized that activated Nrf2 directly regulates Mef2 gene expression in cardiomyocytes. We conducted ChIP followed by quantitative PCR (ChIP-qPCR) of TAC cardiomyocytes to show the significant enrichment of Nrf2 at the Mef2a promoter (Fig. 6f and Supplementary Data 8), validating our hypothesis that the p53–Nrf2–Mef2a axis is essential for the induction of failing cardiomyocytes.

We investigated whether p53 regulates morphological remodeling into heart failure. Pressure overload initially induced cardiomyocyte hypertrophy (Fig. 1b), whereas sustained exposure caused cardiomyocyte elongation (Fig. 6g), consistent with the morphological characteristics of cardiomyocytes from patients with heart failure[45]. However, cardiomyocytes from p53CKO mice maintained the morphological characteristics of hypertrophy and did not show elongation even after chronic pressure overload (Fig. 6h). We also measured the cardiomyocyte cross-sectional area of the heart from p53[flox/flox] and p53CKO mice after sham and TAC operation (Supplementary Fig. 15 and Supplementary Data 9), and obtained results consistent with those of the echocardiographic (Fig. 5b) and single-cardiomyocyte morphological (Fig. 6h) assessments. We conclude that p53 activation in hypertrophy-stage cardiomyocytes is essential not only for molecular remodeling but also for morphological remodeling into failing cardiomyocytes.

**Distinct gene programs and their pathogenicity in human cardiomyocytes.** We analyzed the gene programs and their pathogenicity in human cardiomyocytes. We isolated cardiomyocytes from normal subjects and patients with dilated cardiomyopathy (DCM) ($n = 10$) and obtained 411 single-cardiomyocyte transcriptomes. Among the 17 modules detected by WGCNA (Supplementary Data 10), hierarchical clustering identified five essential modules shared by many cardiomyocytes (Supplementary Fig. 16a). Co-expression network analysis revealed the highly clustered structure of module genes (Fig. 7a) and GO analysis identified specific functions of genes in each module (Fig. 7b). t-SNE and hierarchical clustering analyses separated normal and DCM cardiomyocytes and distinguished five cell clusters (Fig. 7c and Supplementary Fig. 16b). Module expression was specific for cell clusters; M1 was specific for C4 and C5, and M2 was specific for C3 (Fig. 7d, e and Supplementary Fig. 16b). RNA in situ hybridization analysis confirmed the heterogeneous expression of *MYL2* (M2; Fig. 7e and Supplementary Fig. 16c) in DCM cardiomyocytes (Supplementary Fig. 16d). These results uncover the transcriptional heterogeneity and distinct gene modules in human DCM cardiomyocytes.

*CDKN1A*, which is a p53 target gene and belongs to M1 (Fig. 7e and Supplementary Fig. 16c), was highly expressed in a subpopulation of DCM cardiomyocytes (C4 and C5; Fig. 7c, f). smFISH validated an increase of *CDKN1A* expression in some cardiomyocytes (Fig. 7g, h). Co-expression network analysis clarified that *CDKN1A* was strongly correlated with *MT2A* in the M1 network (Fig. 7i). *MT2A* was highly expressed in *CDKN1A*-positive cardiomyocytes (Fig. 7f). Metallothioneins, encoded by MT family genes, are activated by oxidative stress[46] and exert cardioprotection[47], suggesting that *CDKN1A* expression in human cardiomyocytes is induced by oxidative stress.

We also revealed that not only cardiomyocytes from normal subjects but also those from responders, who showed an improvement of heart function after mechanical unloading, showed low M1 and M2 expression (Fig. 7j and Supplementary Fig. 16e). Comparison between human and mouse cardiomyocyte modules confirmed that human M1 and M2 significantly overlapped with mouse M1, M2, M3, M5, M11, and M16 (Fig. 7k). These results demonstrate the conservation of the pathogenic gene programs between human and mouse cardiomyocytes.

## Discussion

Having established a system for the comprehensive analysis of multilayer cardiomyocyte responses to hemodynamic overload in vivo at the single-cell level, we elucidated the trajectory of cardiomyocyte remodeling in response to pathological stimuli, distinguished gene modules for cardiomyocyte hypertrophy and failure, and revealed the coordinated molecular and morphological dynamics of cardiomyocytes leading to heart failure (Fig. 6i).

Upon pressure overload, cardiomyocytes activated the mitochondrial ribosome and oxidative phosphorylation module, the

---

**Fig. 5** p53 is necessary for the emergence of failing cardiomyocytes and the development of heart failure. **a** Representative images of an echocardiographic assessment of p53[flox/flox] (p53[f/f]) and p53CKO mice before and after TAC (weeks 2 and 4). **b** Bar plots showing body weight, cardiac size, and cardiac function evaluated by echocardiography in p53[f/f] and p53CKO mice before and after pressure overload. Mean and standard error of the mean are shown ($n = 12$ [p53[f/f]] and 15 [p53CKO] for pre-TAC, 11 [p53[f/f]] and 10 [p53CKO] for post-TAC W2, and 7 [p53[f/f]] and 7 [p53CKO] for post-TAC W4). Asterisks indicate statistical significance ($P < 0.05$, two-tailed unpaired $t$-test). **c** Violin plot showing the distribution of the correlation coefficients of single-cell transcriptomes among cardiomyocytes from p53[f/f] and p53CKO mice at 2 weeks after TAC (43 p53[flox/flox] and 34 p53CKO cardiomyocytes). **d** Unsupervised hierarchical clustering classifying cardiomyocytes (p53[f/f] and p53CKO cardiomyocytes [TAC W2]) into four cell clusters (cell clusters A–D). Colored bars below the heatmap indicate the cell sources (p53[f/f] [red] or p53CKO [cyan]) and the cell clusters (A–D). **e** t-SNE plots of cardiomyocytes from p53[f/f] and p53CKO mice at 2 weeks after TAC. Cells (dots) are colored by the cell sources (left) and according to the cell clusters in **d** (right). **f** t-SNE plots colored by the expression of each module. **g** Scatter plots showing the expression of M1 and M3 (left) and M3 and M24 (right) in cardiomyocytes from p53[f/f] and p53CKO mice. **h** Bar plot showing the distribution of allocated cardiomyocytes from p53[f/f] and p53CKO mice (TAC W2). **i** Violin plot showing the distribution of the correlation coefficients of single-cell transcriptomes among cardiomyocytes from p53[f/f] and p53CKO mice (TAC W2)

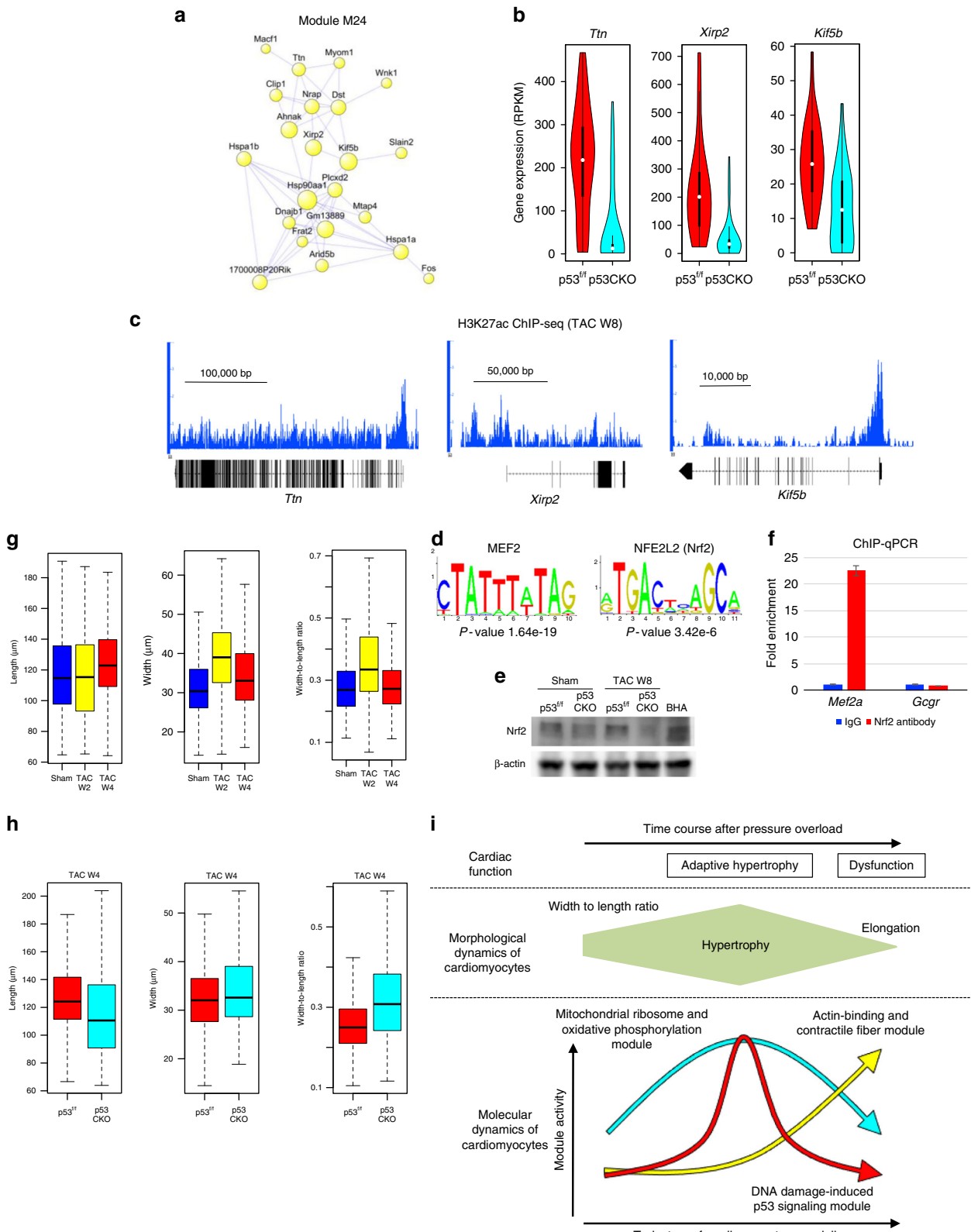

activity of which is correlated with the extent of morphological hypertrophy and is linked to the ELK1 and NRF1/2 transcriptional networks. This suggests that ERK1/2-induced ELK1 phosphorylation[26,27] and NRF1/2 activation[28,29] cooperatively regulate mitochondrial biogenesis to generate more ATP, which is required for enhanced protein synthesis to adapt to the enlargement of cell size and for increased contractility to overcome

pressure overload. We also revealed the association of morphological hypertrophy and the oxidative stress response, supporting the hypothesis that increased mitochondrial biogenesis is the source of oxidative stress[35].

Sustained stimuli induce the accumulation of oxidative DNA damage, leading to p53 signaling activation during hypertrophy. Single-cell analysis of cardiomyocyte-specific knockout mice

**Fig. 6** p53 induces molecular and morphological remodeling leading to heart failure. **a** Co-expression network analysis of M24. **b** Violin plots showing the expression of *Ttn*, *Xirp2*, and *Kif5b* in cardiomyocytes from p53^f/f and p53CKO mice at 2 weeks after TAC (43 p53^flox/flox and 34 p53CKO cardiomyocytes). **c** Representative genome browser views of H3K27ac ChIP-seq of cardiomyocytes at 8 weeks after TAC. The *Y*-axis indicates reads per million (range, 0–4). **d** The enriched transcription factor recognition motifs at the H3K27ac-positive regions (TAC W8 cardiomyocytes) around the M24 genes. **e** Western blot analysis of heart tissues using antibodies against Nrf2 and β-actin. Butylated hydroxyanisole (BHA), an oxidative stress inducer, was used as a positive control. Uncropped images of the blots are shown in Supplementary Fig. 14d. **f** ChIP-qPCR analysis of cardiomyocytes from mice at 8 weeks after TAC operation. Data are represented as mean and standard error of the mean (n = 3 each). The *Gcgr* locus was used as a control region. **g** Boxplots showing the distribution of the morphological parameters of cardiomyocytes from mice after sham and TAC operation (n = 1243 [Sham], 1366 [TAC W2], 717 [TAC W4] from three mice each). Horizontal lines indicate the medians. Boxes show the 25th–75th percentiles. Whiskers represent the minimum and maximum values. **h** Boxplots showing the distribution of morphological parameters of cardiomyocytes from p53^f/f and p53CKO mice at 4 weeks after TAC (n = 1761 [p53^f/f], 1538 [p53CKO] from three mice each). Horizontal lines indicate the medians. Boxes show the 25th–75th percentiles. Whiskers represent the minimum and maximum values. **i** Model for the coordinated molecular and morphological dynamics of cardiomyocytes leading to heart failure

provided strong evidence indicating that p53 in cardiomyocytes increases cell-to-cell transcriptional heterogeneity, induces morphological elongation, and drives pathogenic gene programs by disrupting the adaptive hypertrophy modules and activating the heart failure module, thereby elucidating how DNA damage accumulation leads to heart failure[33,34]. Heterogeneous p53 activation in vivo may be related to recent live-cell imaging results showing stochastic p53 signaling activation for cell fate determination in vitro[48]. Our single-cell analysis uncovered a small population of hypertrophy-stage cardiomyocytes with transient activation of p53 signaling. Heart failure stage-specific cells dominate in the failing heart, suggesting that p53 signaling might be activated in most cardiomyocytes to induce heart failure. This p53 signaling-induced cellular dysfunction might be associated with findings that p53 activation at the G2–M phase is necessary and sufficient for cellular senescence[49,50]. The p53-induced heart failure gene program is orchestrated by MEF2 and Nrf2. We showed that p53 is necessary for Nrf2 activation in the heart failure stage, consistent with the previous finding of p21-mediated Nrf2 protection against Keap1-mediated ubiquitination[51]. Bardoxolone methyl, an antioxidant Nrf2 activator, increases the risk of heart failure[52], suggesting that the p53-induced gene program might cause heart failure.

Single-cell analysis of samples from DCM patients revealed distinct gene programs and their pathogenicity in human cardiomyocytes. *CDKN1A* and *MT2A* co-expression suggested the pathogenicity of oxidative stress. The conservation of pathogenic gene programs in mouse and human cardiomyocytes provides the potential for assessing and regulating cardiomyocyte remodeling in heart failure. Further studies are needed to uncover the relationships between module dynamics in mouse cardiomyocytes and the pathogenicity of human cardiomyocytes and to develop novel therapeutic approaches for preventing heart failure. Finally, we note that the integration of single-cell transcriptomes with other multidimensional data deepens our understanding of disease pathogenesis and may be applicable to other diseases and biological phenomena.

## Methods

**Data reporting**. No statistical methods were used to predetermine sample size. The experiments were not randomized. The investigators were not blinded to allocation during the experiments and outcome assessments.

**Mice**. All animal experiments were approved by the Ethics Committee for Animal Experiments of the University of Tokyo (RAC150001) and Osaka University (22-056) and adhered strictly to the animal experiment guidelines. C57BL/6 mice were purchased from CLEA JAPAN and α-*MHC-Cre*^tg mice were from Jackson Laboratory (stock#009074, stock name: STOCK Tg [Myh6-cre]1Jmk/J). *Trp53*^flox/flox mice[53] were obtained from Dr. Anton Berns (The Netherlands Cancer Institute, Amsterdam, Netherlands). All mice were maintained in specific pathogen-free conditions in the animal facilities of the University of Tokyo and Osaka University. Conditional deletion of *Trp53* in cardiomyocytes was achieved by crossing

*Trp53*^flox/flox homozygous mice with α-*MHC-Cre*^tg hemizygous mice. *Trp53*^flox/flox; α-*MHC-Cre*^tg (*Trp53*^α-MHC-Cre) mice were used as p53 cardiomyocyte-specific knockout (p53CKO) mice. Male mice at 8 weeks of age were used for all experiments. p21 knockout mice were obtained from Dr. Philip Leder (Department of Genetics, Harvard Medical School, Boston, MA)[54]. All mice were on a C57BL/6 background.

**TAC, morphological assessment, and echocardiography**. Mice (8 weeks old) underwent TAC to induce heart failure or sham operation[12,13]. The transverse aorta was constricted with a 7-0 silk suture paralleled with a 27-gauge blunt needle, which was removed after constriction. Sham-operated mice, which were used as controls, had undergone a similar surgical procedure without aortic constriction 2 weeks previously. Transthoracic echocardiography was performed on conscious mice with a Vevo 2100 Imaging System (VisualSonics). M-mode echocardiographic images were obtained from the longitudinal axis to measure left ventricular size and function. The morphological assessment in Figs. 1b and 6g, h was conducted using an Operetta high-content imaging system (Perkin Elmer).

**Single-cell RNA-seq analysis of mouse cardiomyocytes**. Cardiomyocytes were isolated using Langendorff perfusion from the left ventricular free wall after sham operation and at 3 days and 1, 2, 4, and 8 weeks after TAC. Echocardiography was used to assess whether the heart was appropriately exposed to pressure overload. Mice whose hearts were not appropriately exposed to pressure overload were excluded from single-cardiomyocyte RNA-seq analysis. The results of echocardiographic assessment of the mice analyzed in this study are shown in Supplementary Fig. 1a, b. Enzymatic dissociation using Langendorff perfusion was performed with 37 °C pre-warmed 35 mL enzyme solution (collagenase Type II 1 mg/mL [Worthington], protease type XIV 0.05 mg/mL [Sigma-Aldrich], NaCl 130 mM, KCl 5.4 mM, MgCl₂ 0.5 mM, NaH₂PO₄ 0.33 mM, D-glucose 22 mM, HEPES 25 mM, pH 7.4) at a rate of 3 mL/min. Enzymes were neutralized with fetal bovine serum (FBS) at a final concentration of 0.2%. Cell suspensions were filtered through a 100-μm nylon mesh cell strainer and centrifuged at 100 *g* for 2 min. The supernatant was discarded. To prevent hypercontraction, the cardiomyocytes were resuspended in medium (NaCl 130 mM, KCl 5.4 mM, MgCl₂ 0.5 mM, NaH₂PO₄ 0.33 mM, D-glucose 22 mM, HEPES 25 mM, FBS 0.2%, pH 7.4) containing a low concentration of calcium (0.1 mM). Rod-shaped live cardiomyocytes (viability of cardiomyocytes was ≥80% for all time points) were collected immediately after isolation from two mice at each time point with a 0.2–2-μL pipette (sample volume, 0.5 μL) and incubated in lysis buffer. Single-cell cDNA libraries were generated using the Smart-seq2 protocol[15] and the efficiency of reverse transcription was assessed by examining the cycle threshold (Ct) values of control genes (*Tnnt2*, *Cox6a2*) from quantitative real-time polymerase chain reaction (qRT-PCR) using a CFX96 Real-Time PCR Detection System (Bio-Rad) and by examining the distribution of the lengths of cDNA fragments using a LabChip GX (Perkin Elmer) and/or TapeStation 2200 (Agilent Technologies). The following primer sets were used for qRT-PCR: *Tnnt2* mRNA forward, TCCTGGCAGA GAGGAGGAAG; *Tnnt2* mRNA reverse, TGCAGGTCGA ACTTCTCAGC. *Cox6a2* mRNA forward, CGTAGCCCTC TGCTCCCTTA; *Cox6a2* mRNA reverse, GGATGCGGAG GTGGTGATAC. A Ct value of 25 was set as the threshold. The remaining libraries were subjected to paired-end 51-bp RNA sequencing on a HiSeq 2500 in rapid mode. Insert size was 345 ± 40 bp (average ± standard deviation). The data regarding the major statistics and sequencing quality control are summarized in Supplementary Data 1. The RefSeq transcripts (coding and non-coding) were downloaded from the UCSC genome browser (http://genome.ucsc.edu). Reads were mapped to the mouse genome (mm9) using Bowtie (version 1.1.1)[55] with the parameters "-S -m 1 -l 36 -n 2 mm9." RPKM was calculated with reads mapped to the nuclear genome using DEGseq (version 1.8.0)[56]. More than 5000 genes were detected with RPKM > 0.1 in 82.1% of cardiomyocytes (396/482 cells), and more than 4000 genes were detected with RPKM > 1 in 86.9% of cardiomyocytes (419/482 cells) (Supplementary Fig. 1j). Expression levels were comparable between

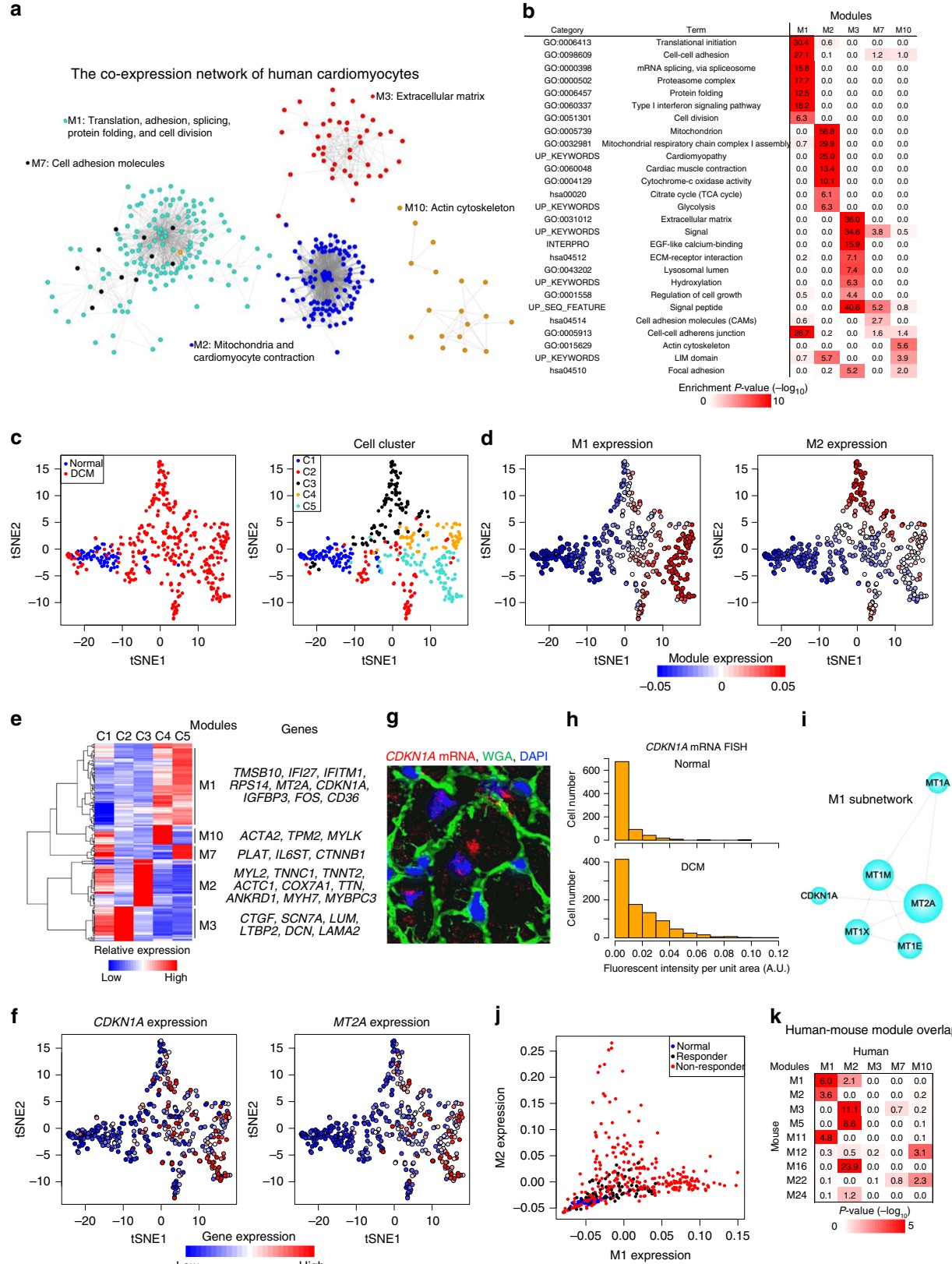

cardiomyocytes that had more than 5000 genes with RPKM > 0.1 and those that had more than 4000 genes with RPKM > 1 (Supplementary Fig. 1k). Among all of the cardiomyocytes that had more than 5000 genes with RPKM > 0.1, over 4000 genes were detected with RPKM > 1 (Supplementary Fig. 1l, m). Reads were also mapped using Tophat[57] with the parameters "-g 1 -p 8 mm9—no-coverage-search"

and the calculated RPKM values and gene modules were compared with those obtained using Bowtie (Supplementary Figs. 1h, i, n and 6a). Two thresholds were used for selecting the cells and genes for downstream analysis; RPKM > 0.1 was used for selecting the cells, and RPKM > 10 was used for selecting the genes. To assess the possibility of batch effects, we added ArrayControl RNA Spikes 1, 4, and

**Fig. 7** Distinct gene programs and their pathogenicity in human cardiomyocytes. **a** Co-expression network of human cardiomyocytes. Dot colors indicate module colors, matching the colors in **b**. **b** Heatmap showing the enrichment of GO and KEGG pathway terms in each module. **c** t-SNE visualization of human cardiomyocytes (71 normal and 340 dilated cardiomyopathy (DCM) cardiomyocytes). Cells (dots) are colored according to the cell clusters in Supplementary Fig. 16b. **d** t-SNE plots of human cardiomyocytes colored by the expression of each module. **e** Heatmap showing the relative average expression for characteristic genes across the five modules. Representative genes are also indicated. **f** t-SNE plots of human cardiomyocytes colored by the expression of each gene. **g** Representative images of *CDKN1A* mRNA smFISH of the DCM heart. WGA and DAPI are used as markers of the plasma membrane and nucleus, respectively. Scale bar, 20 μm. **h** Bar plots showing the distribution of cells corresponding to the single-cell fluorescent intensity of *CDKN1A* mRNA detected by smFISH in the normal and DCM hearts. **i** Subnetwork analysis of human M1. **j** Scatter plot showing the relationship between M1 and M2 expression. DCM cardiomyocytes are separated into two groups: responder ($n = 59$) and non-responder ($n = 281$). **k** Heatmap showing the significance of gene overlaps between human and mouse modules. The table is colored by $-\log_{10}(P\text{-value})$, obtained with Fisher's exact test, according to the color legend below the table

7 (Ambion, cat. No. Am1780) to the lysis buffer at the pre-defined concentrations on two separate plates and conducted single-cardiomyocyte RNA-seq of wild-type mice, confirming a good correlation between RNA spike-in concentrations and their expected RPKM values (normalized by unique reads mapped to the nuclear genome) in both batches (Supplementary Fig. 1c). t-SNE analysis of single-cardiomyocyte transcriptomes from normal C57BL/6 mice (RPKM values) in two different batches confirmed that cardiomyocytes could not be classified by batch (Supplementary Fig. 1d). On the basis of these findings, we considered that we did not need to control for batch effects and used RPKM normalization for quantitative gene expression analysis in this study.

To assess transcriptional heterogeneity between cells, all genes that were expressed at an RPKM value of ≥10 in at least 20% of the samples were used. Pearson's correlation coefficient was calculated using the Cor function in R. Violin plots were generated with the Vioplot package in R.

For weighted co-expression network analysis, all genes expressed at an RPKM value of ≥10 in at least one of the samples ($n = 11{,}479$) were used to construct a signed network using the WGCNA R package[10]. The soft power threshold was analyzed with the "pickSoftThreshold" function and was applied to construct a signed network and calculate module eigengene expression using the "blockwiseModules" function. Modules with <30 genes were merged to their closest larger neighboring module. To visualize the weighted co-expression networks, Cytoscape (version 3.3.0)[58] with the "prefuse force-directed layout" was used. Node centrality, defined as the sum of within-cluster connectivity measures, reflected node size. To calculate the module expression data shown in Figs. 3 and 5, the "moduleEigengenes" function was applied to all cardiomyocytes, including the 396 cardiomyocytes that had been used to obtain the co-expression network modules. Assessment of module overlap was performed using Fisher's exact test with the WGCNA "overlapTable" function[59].

Cluster 3.0 (ref. [60]) and JAVA Treeview[61] were used to perform hierarchical clustering with correlation distance and complete linkage. Minimum cluster size was set to 5% of all samples. To assess the accuracy of the classification, the "randomForest" package in R was used. After obtaining nine modules by the combination of random forests and hierarchical clustering, we performed the subsequent cell clustering analysis based on the PC1 values of the nine modules to classify cardiomyocytes. This is because the combination of random forests and hierarchical clustering showed that the accuracy of classification using the PC1 values of the nine modules was higher than that using the gene expression profiles of the nine modules (Supplementary Fig. 7a–c). To achieve accurate cell-type classification, in Fig. 1d, we compared the error rates for 4–10 clusters and found that they were maintained at a low level for 4–7 clusters, while a drastic increase of the error rate was observed for 8 clusters (Supplementary Fig. 7b). A comparison of the hierarchical clustering results using the gene expression profiles with those from using the PC1 values showed that most of the cells classified into the same cell clusters were well clustered (Supplementary Fig. 7a). To visualize cell-to-cell variations, the expression of modules identified to be significant for cell classification by the Random Forests algorithm[17] were applied to the t-SNE algorithm[18] (perplexity = 10) with the "Rtsne" package in R in Figs. 1g and 5e and Supplementary Fig. 9. Principal component analysis was performed using the "prcomp" function in R.

The Database for Annotation, Visualization, and Integrated Discovery (DAVID)[62] was used for GO analysis and Kyoto Encyclopedia of Genes and Genomes (KEGG) pathway enrichment analysis. For Figs. 1f and 4f, the most characteristic GO terms that ranked in the top 3 in the "Functional Annotation Clustering" function with statistical significance ($P < 0.05$) were extracted for each module. Enrichment $P$-values of all extracted GO terms for each module were calculated in DAVID. Monocle[19,20] was used for differential expression analysis ($q$-values <1e-6), trajectory analysis, and pseudo-time analysis.

For the integrative analysis of single-cell morphology and transcriptomes shown in Fig. 3, images of 48 cardiomyocytes from two mice at 1 week after TAC were obtained using an inverted microscope with a digital camera (CKX31 and XZ-2; Olympus) soon after isolation by Langendorff perfusion. Cardiomyocytes were then incubated immediately in lysis buffer. Subsequent procedures for extracting single-cardiomyocyte transcriptomes were the same as those described above. After removing seven libraries showing incomplete reverse transcription, we sequenced 41 libraries and obtained 36 single-cardiomyocyte transcriptomes that included

over 4500 genes with RPKM > 0.1, which were used for integrative analysis of single-cell morphology and transcriptomes. Cardiomyocyte size was analyzed using NIH ImageJ software (version 1.48)[63].

To assess the possibility of morphological changes after isolation in Supplementary Fig. 11, images of single cardiomyocytes were obtained in the medium at 0, 1, 5, 30, and 60 min after isolation using a BZ-X700 microscope (Keyence). The data for area, length, width, and length-to-width ratio were analyzed using NIH ImageJ software (version 1.48)[63].

Cardiomyocytes from wild type and p53CKO mice (48 wild type and 48 p53CKO cardiomyocytes) were also isolated using Langendorff perfusion at 2 weeks after TAC from two wild type (p53[f/f]) and two p53CKO mice and then incubated immediately in lysis buffer. The subsequent procedures for extracting single-cardiomyocyte transcriptomes were the same as those described above. To quantify *Trp53* mRNA levels with single-cardiomyocyte qRT-PCR, the following primer set was used: forward, GACGGGACAG CTTTGAGGTT; reverse, GCAGTTCAGG GCAAAGGACT. Delta Ct values were calculated by subtracting *Trp53* mRNA Ct values from the threshold cycle number of 35 (Supplementary Fig. 13f). Cardiomyocytes (43 wild type and 34 p53CKO) whose transcriptomes included over 3000 genes with RPKM > 0.1 were used for the subsequent analysis. To allocate the wild type and p53CKO cardiomyocytes into the predetermined clusters, the "randomForest" package in R was used.

**ChIP-qPCR and ChIP-seq analysis.** Cardiomyocytes ($1.0 \times 10^6$ cells) were isolated from two mice at 1 week after TAC with Langendorff perfusion, crosslinked with 1% formaldehyde at room temperature for 10 min, and quenched with 0.125 M glycine. After sonication, soluble chromatin was added to 4 μL recombinant histone H2B (New England BioLabs) and 0.5 μL mouse RNA (QIAGEN) and incubated with an antibody-bead complex (5 μg anti-H3K27ac antibody [ab4729; Abcam] or 5 μg anti-Nrf2 antibody [61599; Active motif] and 20 μg Dynabeads Protein G [10003D; Thermo Fisher Scientific]) at 4 °C overnight. Immunoprecipitates were washed, reverse-crosslinked, and incubated with proteinase K and RNase A. DNA was purified using a MinElute PCR Purification Kit (Qiagen). Extracted DNA was used for qPCR and sequencing. Fold enrichment was determined as the fold change in percent input (ChIP signal/input signal) at the target regions compared with the control region. Rabbit IgG was used as a negative control. Primer sequences were as follows: *Mef2a* forward, TGTGGCACAA GGACAAGAGG; *Mef2a* reverse, TCCAGGAGAA CACGGTCCTT; *Gcgr* forward, TGCTGTCATG TCTGGTGAGTG; *Gcgr* reverse, GGAGCTGTCA GCACTTGTGTA. Sequencing libraries were prepared using the TruSeq ChIP Library Preparation Kit (Illumina) for high-throughput sequencing on an Illumina HiSeq 2500 according to the manufacturer's protocol.

Reads were mapped to the mouse genome (mm9) using Bowtie (version 1.1.1)[55] with the parameters "-S -m 1 -l 36 -n 2 mm9." ChIP-seq peaks were identified with the MACS program (version 1.3.7.1)[64] with a $P$-value <1e-5. The other parameters were as follows: --tsize "36," --gsize "1865500000," --mfold "10," --nomodel,-- nolambda, --wig. Sequence reads aligning to the region were extended by 300 bp and the density of reads in each region was normalized to the total number of million mapped reads to produce the normalized read density in units of reads per million mapped reads. The Integrated Genome Browser[65] was used to visualize the ChIP-seq peaks. H3K27ac-positive regions whose nearest genes were assigned to specific modules were defined as putative regulatory elements for each module. For motif analysis, these elements were analyzed using the TRanscription factor Affinity Prediction (TRAP) algorithm[66,67] (weight score threshold = 5). TRANSFAC[68] and JASPAR[69] were used as the databases. Mouse non-coding sequences were used as background for the $P$-value calculations. $P$-values corrected with the Benjamini–Hochberg procedure are shown. Cluster 3.0 (ref. [60]) and JAVA Treeview[61] were used to perform hierarchical clustering of transcription factor recognition motifs.

**Single-molecule RNA fluorescent in situ hybridization.** For single-molecule RNA fluorescent in situ hybridization, the RNAscope system[16] (Advanced Cell Diagnostics) was used. Probes against mouse *Myh7* mRNA (NM_080728.2, bp2- 6054, ACD# 454741), mouse *Atp2a2* mRNA (NM_001110140.3, bp535-2533, ACD# 454731), mouse *Cdkn1a* mRNA (NM_007669.4, bp215-1220, ACD#

408551-C2), mouse *Pecam1* mRNA (NM_001032378, bp915-1827, ACD#316721), mouse *Cav1* mRNA (NM_007616.4, bp140–1290, ACD#446031), mouse *Lum* mRNA (NM_008524.2, bp254–1336, ACD#480361), mouse *Dcn* mRNA (NM_007833.5, bp2–1207, ACD#413281), and human *CDKN1A* mRNA (NM_078467.2, bp74–1500, ACD# 311401-C2) were used. Frozen sections (10 μm) were fixed in phosphate-buffered saline (PBS) containing 4% paraformaldehyde for 15 min at 4 °C, dehydrated by serial immersion in 50%, 70%, and 100% ethanol for 5 min at room temperature, and protease-treated for 30 min at room temperature. The probes were then hybridized for 2 h at 40 °C, followed by RNAscope amplification, and co-stained with Alexa Fluor 488-conjugated wheat germ agglutinin (WGA; W11261; Thermo Fisher Scientific; 1:100) to detect cell borders. Sections were counterstained with DAPI. Images were obtained as Z stacks using an LSM 510 META confocal microscope (Zeiss) and/or In Cell Analyzer 6000 (GE Healthcare). Quantification of fluorescence intensity was performed using the In Cell Developer Toolbox (GE Healthcare). We defined individual cardiomyocytes using WGA staining, measured fluorescence intensity within each cardiomyocyte, and normalized this intensity by cell area, quantifying the single-cell fluorescence intensity per unit area.

**Immunohistochemistry.** Fresh sections (5 μm) were fixed in PBS containing 4% paraformaldehyde for 10 min at room temperature and rinsed three times with PBS for 5 min. The sections were blocked with the blocking reagent supplied in a MOM Immunodetection Kit (Vector Laboratories) containing 5% normal goat serum for 1 h at 4 °C and rinsed three times with PBS for 5 min. The sections were incubated with the following antibodies overnight at 4 °C: mouse monoclonal anti-p21 antibody conjugated to Alexa Fluor 647 (1:50; sc6246 AF647; Santa Cruz Biotechnology) and rabbit polyclonal anti-γH2A.X (phospho S139) antibody (1:100; ab2893; Abcam). For γH2A.X staining, a secondary antibody (goat anti-rabbit IgG [H + L] conjugated to Alexa Fluor 546 [1:200; A11010; Thermo Fisher Scientific]) was used. Alexa Fluor 488-conjugated WGA (1:100) was applied to visualize the plasma membrane. Sections were mounted with ProLong Gold Antifade with DAPI (Thermo Fisher Scientific). Images were obtained using an LSM 510 META confocal microscope (Zeiss) and/or a BZ-X700 microscope (Keyence).

**Cardiomyocyte cross-sectional area measurement.** For immunofluorescent staining, hearts were excised and embedded immediately in Tissue-Tek OCT compound (Sakura Finetek Inc.). Cryosections at 5 μm were fixed in PBS containing 4% paraformaldehyde for 15 min at 4 °C. The sections were blocked in 5% normal goat serum. The sections were counterstained with Alexa Fluor 488-conjugated WGA (W11261; Thermo Fisher Scientific; 1:100) and mounted with ProLong Gold Antifade with DAPI (Life Technologies). Images were obtained using a BZ-X700 microscope (Keyence). Cardiomyocyte cross-sectional area was measured using a BZ analyzer (Keyence).

**Western blot analysis.** Heart tissues were homogenized and lysed with RIPA buffer containing 10 mM Tris-HCl, 150 mM NaCl, 5 mM EDTA, 1% Triton X-100, 1% sodium deoxycholate, and 0.1% sodium dodecyl sulfate with protease and phosphatase inhibitor cocktails. Lysates were centrifuged at 20,000 *g* for 40 min at 4 °C and the supernatants were recovered. Total protein concentrations in the supernatants were measured using a bicinchoninic acid assay (Pierce BCA Protein Assay Kit; Thermo Scientific). For immunoblot analysis, extracted protein samples were separated on 7.5% Mini-PROTEAN TGX precast gradient gels (Bio-Rad) and transferred onto nitrocellulose membranes. The membranes were blocked with 5% FBS in Tris-buffered saline plus 0.05% Tween and incubated overnight at 4 °C with an anti-NRF2 antibody (1:1000; Active Motif) and an anti-actin antibody (1:5000; Thermo Fisher Scientific) as a loading control. Primary antibodies were detected with horseradish peroxidase-conjugated species-specific secondary antibodies (GE Healthcare) and ECL plus (Thermo Fisher Scientific) using a LAS 3000 analyzer (GE Healthcare). Immunoblot band intensities were measured using NIH ImageJ software[63]. Butylated hydroxyanisole (BHA; Sigma-Aldrich) was administered intraperitoneally to male mice at 8 weeks of age at a dose of 350 mg/kg in corn oil. Uncropped scans of blots are shown in Supplementary Fig. 14a, b, d.

**Single-cell RNA-seq analysis of human cardiomyocytes.** All experiments were approved by the ethics committee of the University of Tokyo (G-10032). All procedures were conducted according to the Declaration of Helsinki, and all patients gave written informed consent before taking part in the study. Heart samples from normal subjects and patients with DCM (*n* = 10) were used. Heart tissues were obtained immediately after death due to non-cardiac causes (normal cardiac function) or during left ventricular assist device (LVAD) surgery, respectively. Responders were classified by serial post-LVAD echocardiography as a relative increase in left ventricular ejection fraction >50%.

Immediately after the collection of the heart tissue, rod-shaped live cardiomyocytes were isolated[70], and then incubated in lysis buffer. Single-cell cDNA libraries were generated using the Smart-seq2 protocol[15], and the efficiency of reverse transcription was assessed by examining the cycle threshold (Ct) values of a control gene (*TNNT2*) from qRT-PCR using a CFX96 Real-Time PCR Detection System (Bio-Rad) and by examining the distribution of the lengths of cDNA fragments using a LabChip GX (Perkin Elmer) and/or TapeStation 2200

(Agilent Technologies). The following primer set was used for qRT-PCR: *TNNT2* mRNA forward, AAGTGGGAAG AGGCAGACTGA; *TNNT2* mRNA reverse, GTCAATGGCC AGCACCTTC. A Ct value of 25 was set as the threshold. The remaining libraries were sequenced using a HiSeq 2500 System (Illumina). The data regarding the major stats and sequencing quality control are summarized in Supplementary Data 2. Reads were mapped to the human genome (hg19) using Bowtie (version 1.1.1)[55] with the parameters "-S -m 1 -l 36 -n 2 hg19." RPKM values were calculated with reads mapped to the nuclear genome using DEGseq (version 1.8.0)[56]. Single-cell transcriptomes in which >5000 genes were detected (RPKM > 0.1) (*n* = 411) were used for subsequent analysis.

For weighted co-expression network analysis, all genes expressed at an RPKM value of ≥5 in at least 40 samples (*n* = 7815) were used to construct a signed network using the WGCNA R package[10]. The soft power threshold was analyzed with the "pickSoftThreshold" function and was applied to construct a signed network and calculate module eigengene expression using the "blockwiseModules" function. Modules with <30 genes were merged to their closest larger neighboring module. To visualize the weighted co-expression networks, Cytoscape (version 3.3.0)[58] with the "prefuse force-directed layout" function was used. Node centrality, defined as the sum of within-cluster connectivity measures, reflected node size. The "overlapTable" function was used to calculate the mouse and human module overlaps.

Cluster 3.0 (ref. [60]) and JAVA Treeview[61] were used to perform hierarchical clustering. To visualize cell-to-cell variations, the expression of modules was applied to the t-SNE algorithm[18] with the "Rtsne" package in R. DAVID[62] was used for GO analysis and KEGG pathway enrichment analysis. The most characteristic GO terms in annotation clusters that ranked in the top 7 in the "Functional Annotation Clustering" function with statistical significance (*P* < 0.05) were extracted for each module. Enrichment *P*-values of all extracted GO terms for each module were calculated in DAVID.

**RNA in situ hybridization of human samples.** Human tissues were fixed with G-Fix (Genostaff), embedded in paraffin on CT-Pro20 (Genostaff) using G-Nox (Genostaff) as a less toxic organic solvent than xylene, and sectioned at 5 μm. RNA in situ hybridization was performed with an ISH Reagent Kit (Genostaff) according to the manufacturer's instructions. Tissue sections were de-paraffinized with G-Nox, and rehydrated through an ethanol series and PBS. The sections were fixed with 10% NBF (10% formalin in PBS) for 30 min at 37 °C, washed in distilled water, placed in 0.2 N HCl for 10 min at 37 °C, washed in PBS, treated with 4 μg/mL proteinase K (Wako Pure Chemical Industries) in PBS for 10 min at 37 °C, washed in PBS, and then placed within a Coplin jar containing 1× G-Wash (Genostaff), equal to 1× SSC. Hybridization was performed with probes (250 ng/mL) in G-Hybo-L (Genostaff) for 16 h at 60 °C. After hybridization, the sections were washed in 1× G-Wash for 10 min at 60 °C and in 50% formamide in 1× G-Wash for 10 min at 60 °C. Then, the sections were washed twice in 1× G-Wash for 10 min at 60 °C, twice in 0.1× G-Wash for 10 min at 60 °C, and twice in TBST (0.1% Tween 20 in TBS) at room temperature. After treatment with 1× G-Block (Genostaff) for 15 min at room temperature, the sections were incubated with anti-DIG AP conjugate (Roche Diagnostics) diluted 1:2000 with G-Block (Genostaff; dilated 1/50) in TBST for 1 h at room temperature. The sections were washed twice in TBST and then incubated in 100 mM NaCl, 50 mM MgCl₂, 0.1% Tween 20, and 100 mM Tris-HCl (pH 9.5). Coloring reactions were performed with NBT/BCIP solution (Sigma-Aldrich) overnight and then washed in PBS. The sections were counterstained with Kernechtrot stain solution (Muto Pure Chemicals) and mounted with G-Mount (Genostaff).

**Statistical analysis.** Normality of data distribution were assessed with the Shapiro–Wilk test. Statistically significant differences between two groups for continuous variables were determined by a two-tailed unpaired *t*-test with an F-test to confirm the equality of variance. Statistically significant differences in allocated clusters or genes between groups were assessed with Fisher's exact test. Significance was considered at *P* < 0.05 unless otherwise indicated.

## Data availability
The data that support the findings of this study are available from the authors on reasonable request. The sequences of the primers used in this study is listed in Supplementary Data 11. Single-cardiomyocyte RNA-seq and cardiomyocyte ChIP-seq data in this study have been deposited in the Gene Expression Omnibus with the accession code GSE95143.

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

## Acknowledgements

We thank K. Shiina for next-generation sequencing support and Y. Yokota for experimental support. This work was supported by grants from the Japan Foundation for Applied Enzymology (to S.N.), the SENSHIN Medical Research Foundation (to S.N.), the KANAE Foundation for the Promotion of Medical Science (to S.N.), MSD Life Science Foundation (to S.N.), The Tokyo Biomedical Research Foundation (to S.N.), Astellas Foundation for Research on Metabolic Disorders (to S.N.), The NOVARTIS Foundation (Japan) for the Promotion of Science (to S.N.), the Japanese Circulation Society (to S.N.), a Grant-in-Aid for Young Scientists (B) (to S.N.), a Grant-in-Aid for Research Activity Start-up (to M.S.), the Practical Research Project for Rare/Intractable Diseases from the Japanese Agency for Medical Research and Development (to H. Aburatani and I.K.), a Grant-in-Aid for Scientific Research (A) (to I.K.), AMED-PRIME, AMED (JP18gm6210010) (to S.N.), and AMED-CREST, AMED (JP18bm0804010, JP18gm0810013) (to H. Aburatani and I.K.).

## Author Contributions

S.N., H. Aburatani, and I.K. conceived the project, designed the study, and interpreted the results; S.N. and M.S. collected single cells and generated the single-cell sequencing data; S.N. performed computational analyses; T.F. and H. Aburatani provided support for computational analyses; T.H. generated p53CKO mice, performed TAC, and conducted the functional analysis; T.S. and T.Y. performed TAC and conducted the functional analysis; S.N. and M.S. performed biochemical experiments and analyzed the data; T.K. supported the immunohistochemistry experiments; T.T., A.T.N., M.I., K.F., M.H., H.T., Y.K., K.I., E.T., H. Akazawa, H.M., H. Aburatani, and I.K. provided experimental and analytical support; S.N. and I.K. wrote the manuscript with feedback from all authors.

## Additional information

**Competing interests:** The authors declare no competing interests.

