## [Peer Review File · Nature Communications]

Reviewers' comments:

Reviewer #1 (Remarks to the Author):

This work aimed to integrate a large amount of single-cell RNA seq, cell morphology estimates, epigenomic states and physiological heart function data to study the molecular and morphological dynamics of cardiomyocytes during hypertrophy and heart failure. The authors collected a rich dataset that was challenging to integrate and analyze. It required extensive data cleaning / filtering, various optimization steps and integration of results coming from various analytical methods. The data could potentially shed light to several aspects of the heart disease dynamics, differences across cell types and help identify common / distinct gene regulators.

Combining single-cell RNA-seq, cell morphology and epigenomic states is important but it is our opinion that the current work did not fully exploit these three sources of information. Much important information was omitted, that could be helpful for the reviewer/reader.

1. On pages 5 and 34 the authors describe the methodology regarding the isolation and sequencing of the 396 cardiomyocytes coming from various time points. I would have liked to see more details on the data filtering that followed after the 540 cardiomyocytes isolation. What were the characteristics of the 86 libraries that were filtered out in the second filtering step? Was the filtering based on parameters other than the single-cell RNA-seq data information (e.g. cell morphology)? How the cut-off of Ct = 25 was obtained in the first-step filtering?
2. It is unclear how many biological replicates of mice were used.
3. The authors performed single cell RNA-seq on Illumina HiSeq2500. However, important details were missed out which will aid in the technical assessment
 - a. Paired-end or single-end
 - b. Read lengths
 - c. Insert size
4. Related to point 1 above, no mention of how the quality of reads was assessed. For example, the authors might have included libraries with average base quality > 30 in the downstream analysis because low base quality results in technical noise and drawing unreliable conclusions. According to the methods section, FPKM 10 was used as a threshold to call expressed genes. But in the results section, RPKM>0.1 was used. This is conflicting information. Even for the use of both FPKM and RPKM: although technically equivalent, it would be good to be consistent. Perhaps to clarify, it would be useful to be clear how FPKM or RPKM were derived. RPKM>0.1 seems low stringency, and we expect to result in significant noise: how do the authors justify this choice?
5. It was also unclear if the authors looked at protein coding genes or non-coding genes or both. Did they derive information from UCSC, Ensembl or GENCODE?
6. We prefer commonly used mapping software such as STAR or Tophat for alignment of RNA-seq reads. Many reads are not mapped by Bowtie1, particularly when reads span 2 exons involving splicing. Did the authors explore options?
7. Major stats and QC reflecting quality of the sequencing runs are not provided. Without these stats, it is not possible to tell if the results are due to technical artifact. Some important metrics are:
 - a. Library sizes
 - b. Total mapped reads
 - c. Mapping efficiency (% uniquely mapped reads)
 - d. % duplication
 - e. % reads aligned to ribosomal RNA
 - f. % reads aligned to mitochondrial DNA
 - g. % reads aligned to introns and intergenic regions
8. On page 5 the authors state that the cells were isolated at different time points and, as a result, it is possible that their transcriptome is influenced by technical factors (batch effects). How did the authors test and, if needed, controlled for these effects? Batch effect is the singular most difficult technical hurdle such biologically targeted projects face. The authors will need to rigorously

demonstrate how they have accounted for this.

9. In single-cell studies it is often suggested the use of spike-in controls for data normalization. This could be especially useful for normalizing for batch effects. Did the authors include such controls and if not why?

10. It is not clear from the analysis of page 5 how many samples were excluded at each time point.

11. In the results section, the authors mention "expressed profiles were tightly correlated between the average expression of single cells and corresponding bulk expression" It was unclear if the authors averaged the single cell profiles across all timepoints? Were there any point estimates for the correlation? Is "between biological replicates" referring to biological replicates of bulk RNA-seq or single cell RNA-seq.

12. It is a shame that the authors did not attempt to provide a global overview of the sham-TAC single cell transcriptomic landscape before they dived into detailed network analysis. This global analysis would have allowed the analysis for:

a. How heterogeneous are sham and TAC cells. Do they cluster well?

b. Pointing out obvious outliers, and / or cell type contaminations (eg fibroblasts, endothelial cells)

c. Correlation with potential confounding factors such as batch effect, number of expressed genes etc.

13. We found the combination of random forests and hierarchical clustering tools of page 6 very interesting. However, there was a lot of information condensed in a single paragraph that makes it difficult for the reader to understand, and thus making it confusing. Expression profiles were used in WGCNA to obtain 55 modules whose PC1 were extracted. The random forests were applied to the PC1 values to select the 9 most important modules. Was the subsequent cell clustering based on the PC1 data of these 9 modules or the gene expression profiles of the genes included in these 9 modules? It is not clear to us what would be the most accurate way to cluster the cells and we think that this depends on the variance explained by the PC1. We would have liked to see a comparison of these two alternatives and the reasoning behind the authors' preferred choice.

14. From supplementary figure 2b it is evident that the 9 modules give 7 random forest based clusters with low error rates. What parameters / algorithm were used to derive these 7 clusters? The random forest is supervised (based on these 7 clusters). It would be interesting to see how the error rate changes for 5-10 clusters and what clusters an unsupervised algorithm would obtain (e.g. clustering / trajectory estimation and states by Monocle).

15. Pages 6-7 discuss the cell trajectory on the tSNE plot but we could not find any details on how this trajectory was estimated. Results based only on visual inspection can be highly misleading due to the data noise, the optimization process and the huge amount of information included in a typical single-cell dataset. Other than that, the analysis of pages 7-8 with the module variability is very interesting but it needs to be supported by a more rigorous statistical analysis.

16. It is not clear how ImageJ measured the cell area. Were the estimates derived from 2D images? How was the volume of the cell assessed? What does "small" and "large" cell means in figure 2b. What is the variability in cell size and how is it correlated with the cell library size? Would spike-in or endogenous controls be a better way to normalize this dataset?

17. How do the authors ensure that cell sizes were maintained following harvest? Our experience suggests that cardiomyocytes are very sensitive to hypercontraction depending on calcium concentrations when isolated out ex vivo. Is there a way to validate this?

18. Page 10 summarizes the logic behind network comparison but it lacks the details of the methodology followed. How were the differences of the hypertrophy / non-hypertrophy networks been statistically assessed? The information of this paragraph, the paragraph of page 12 and the supporting figures was not enough to understand the key steps of this analysis.

19. There was no differential expression analysis to assess the differences across the cardiomyocyte isolation time points, across the estimated clusters and across the cell trajectory. This information is important in order to assess the significance of the findings. It could point out cell contamination (other cell types), which was not discussed in this work.

20. It is not clear what value H3K27ac ChIP-seq added to the understanding of the modules involved in cardiomyocyte hypertrophy.

Despite these, the methodologies appear to have worked well to produce some real biological insights, that have been validated by concordance both within their own data (multiple seq datasets, including mouse and human, and RNA-FISH/immunostaining). Importantly, this has also been consistent with published literature. However this consistency with published literature also means that a lot of their findings are not actually all that surprising and have been reported before (morphological hypertrophy correlating with increases in genes for translation and mitochondrial biogenesis, adaptive hypertrophy followed by maladaptive and heart failure, cellular heterogeneity in hypertrophy, Mef2, ERK and NRF signalling, oxidative stress -> DNA damage -> disease progression). Admittedly, the temporal and single cell resolution of this study offers some advantages, and the datasets will provide a useful resource to researchers, assuming they are made publically available. Still, the authors did not demonstrate or explain how exactly "these results provide a potential application for assessing and regulating cardiomyocyte remodelling in cardiac hypertrophy and failure."

Perhaps this was why the authors pushed the p53 story, the novelty of which is that p53 activation is critically important specifically at the point of transition from an adaptive hypertrophied cardiomyocyte, to a failing one. However this has two key problems.

The first problem is that a PNAS paper came out earlier this year showing that CM-specific KO of p53 makes hearts resistant to TAC (Mak et al 2017: <http://www.pnas.org/content/114/9/2331.full>). Although analysis was not CM-specific, this paper used microarray profiling to show many of the same effects (e.g. p21, Mef2, mitochondrial biogenesis and bioenergetics, glucose and fatty acid metabolism). The authors of the current study have not cited and seem unaware of this paper. Unfortunately it detracts significantly from the novelty.

The second problem is that, with the massive amount of data from single cell analysis, the authors have not present enough evidence to justify the conclusion that transient p53 activation specifically drives the CM transition from compensated to failing. The above paper shows that p53 has physiological activity in mouse CM at all stages of health/disease. Cardioprotective transcriptional changes are present in p53 KO hearts by day 7 TAC – that is before the appearance of failing CM. It is therefore conceivable that p53 KO hearts maintain a generally higher level of health in TAC than wild type hearts do, and it is because of this that CMs in KO hearts never reach the stage of transition to failure, rather than because of a specific lack of p53 activation at that point. Perhaps if left longer than 8 weeks, failing cells would appear in p53 KO hearts.

Aside from the KO model, the authors base their evidence for p53 activation driving the failure transition on a very small number of cells. p53 activation is not specifically measured, only upregulation. It is possible that in these cells, p53 activation/upregulation is an effect of an already damaged and failing cell, rather than the cause of it. The authors make no mention of p53 upregulation in human DCM cells. Was it not detected? If not, how is this discrepancy explained?

Reviewer #2 (Remarks to the Author):

The authors performed exhaustive single-cell RNA-seq to define the changes in RNA expression that occur in response to pressure overload (TAC) of the heart in mice. A time course of TAC was performed, ranging from early stage compensatory hypertrophy (day 3 and week 1) to late stage heart failure (week 8). Distinct gene regulatory modules were defined computationally at each stage of the response. The transition from hypertrophy to failure was marked by induction of p53-dependent gene expression (Module 24 genes). Interestingly, p53 appeared to regulate Module 24 gene expression, at least in part, by stimulating MEF2 and NRF2 transcription factor-dependent gene expression. To follow up on this, the authors generated cardiomyocyte-specific p53 KO mice. These animals appeared to be protected from heart failure induced by TAC. Finally, the authors perform single-cell transcriptomics on human cardiomyocytes from normal controls and hearts from individuals with dilated cardiomyopathy.

The authors are to be commended for performing a tour de force evaluation of the transcriptional networks that are associated with cardiac hypertrophy. The datasets will provide important resources for the field. The segment of the manuscript that is dedicated to p53 is nice because it provides a mechanistic evaluation that deviates from the somewhat encyclopedic nature of the other figures. I suggest that the authors expand on the p53-related findings to enhance the overall manuscript.

Specific points

1. For Fig. 1B, it would be useful to provide length-to-width ratios for myocytes from other time points following TAC.
2. For sFig. 5F, the authors should perform single-cell qPCR for Trp53 from WT mice post- 2 weeks TAC to rule out effects of the floxed allele on p53 expression.
3. The authors should quantify myocyte cross-sectional area in p53 KO hearts vs. controls, before and after TAC.
4. The manuscript would be enhanced by experiments that address the mechanisms by which p53 regulates MEF2 in cardiomyocytes. To my knowledge, such findings would be highly novel.
5. The authors mention prior work showing p53-induced NRF2 activation via p21-mediated NRF2 protection against Keap1-mediated ubiquitination. They should determine if this mechanism is blocked in p53 KO hearts.

Reviewer #3 (Remarks to the Author):

This is a valuable manuscript studying how cardiomyocytes undergo molecular and morphological changes in response to stress, leading to cardiac hypertrophy and failure. The authors relate these changes to gene expression profiles measured using single-cell mRNA-Seq at cardiomyocytes samples collected at different time points representing disease progression.

The authors find interesting relationships between gene modules and trajectories of disease progression. Gene modules represent sets of genes that show similar expression and are significantly expressed across cells. The identified modules include important genes known from literature to be responsible for the cardiac hypertrophy and the heart failure.

Overall, the structure and sequence of the results is well presented and written. It starts by investigating the responsible genes for early stage hypertrophy, then for the progression to heart failure. Based on their data, they conclude p53 being responsible for the heart failure progression, which they validate using a p53 blocking mouse. Finally, they validate these responsible gene modules in human tissues.

The analysis of the single-cell mRNA-Seq data is well structured and is mainly based on the WGCNA R package pipeline that is widely known and used in gene co-expression analysis. The authors combined these analyses with various visualization tools to produce clear figures, as well as function enrichment analysis to interpret the observed correlations between the observed important genes and the disease progression.

In conclusion, I am very happy with this manuscript. I do have a few comments that I would like to see addressed.

Major comments:

1. In the hierarchical clustering of co-expressing genes into gene modules for the data of the C57BL/6 mouse model, the dendrogram in Figure 1c does not support the 7 cluster assignments mentioned. Further clarification is needed if there are any other criteria for clusters assignment.

2. In the analysis of the hypertrophy related modules, Figure 2b shows a scatter plot of PC1 and PC2 with only 34 cells (dots) while the authors previously mentioned that in week one 82 cardiomyocytes are obtained. Why is there a difference?

Minor comments:

1. For the purpose of reproducibility of the analysis, please specify the perplexity used for producing the t-SNE maps, as well as the distance measure and linkage method used for the hierarchical clustering.

2. Page 34, line 5, remove "at 2 weeks". It is already mentioned in line 6.

Conclusion:

I recommend this work to be published after addressing my comments.

Responses to the reviewers' comments

We thank the reviewers for their insightful comments regarding our manuscript. Along the lines suggested by the reviewers, we have performed additional analyses and revised the manuscript. The sentences revised according to the reviewers' comments are highlighted in red in the main text.

Response to Reviewer #1

(Remarks to the Author):

This work aimed to integrate a large amount of single-cell RNA seq, cell morphology estimates, epigenomic states and physiological heart function data to study the molecular and morphological dynamics of cardiomyocytes during hypertrophy and heart failure. The authors collected a rich dataset that was challenging to integrate and analyze. It required extensive data cleaning / filtering, various optimization steps and integration of results coming from various analytical methods. The data could potentially shed light to several aspects of the heart disease dynamics, differences across cell types and help identify common / distinct gene regulators.

Combining single-cell RNA-seq, cell morphology and epigenomic states is important but it is our opinion that the current work did not fully exploit these three sources of information. Much important information was omitted, that could be helpful for the reviewer/reader.

1. On pages 5 and 34 the authors describe the methodology regarding the isolation and sequencing of the 396 cardiomyocytes coming from various time points. I would have liked to see more details on the data filtering that followed after the 540 cardiomyocytes isolation. What were the characteristics of the 86 libraries that were filtered out in the second filtering step? Was the filtering based on parameters other than the single-cell RNA-seq data information (e.g. cell morphology)? How the cut-off of Ct = 25 was obtained in the first-step filtering?

Response

We apologize for not including these details in the original manuscript. We used two

filtering steps in this paper. The first step was based on an assessment of the efficacy of reverse transcription and amplification using real-time quantitative PCR. The second step was based on an assessment of the number of detected genes after sequencing the single-cell libraries. We did not use any other parameters. Regarding morphology, we randomly collected rod-shaped live cardiomyocytes (viability of cardiomyocytes was $\geq 80\%$ for all time points) immediately after their isolation using Langendorff perfusion. A number of samples obtained during the filtering steps are shown in **Supplementary Fig. 1e**.

In the first step, we conducted real-time quantitative PCR analysis of endogenous control genes (*e.g.*, *Tnnt2* and *Cox6a2*) on all single-cell cDNA libraries (**Supplementary Fig. 1f**). Both Ct values were less than 25 in most of the libraries, suggesting the high efficacy of the reverse transcription and amplification processes. Therefore, we set a Ct value of 25 as the threshold. We also used a LabChip GX (Perkin Elmer) to confirm that the libraries with both Ct values < 25 showed high efficacy for the reverse transcription and amplification processes, whereas those with both Ct values > 25 showed low efficacy (**Supplementary Fig. 1g**).

In the second step, we counted the detected genes (RPKM > 0.1) for all cells after sequencing the libraries, mapping the reads, and calculating the RPKM values (**Supplementary Fig. 1i**). At least 5,000 genes were detected in most cardiomyocytes. Therefore, we set a threshold of 5,000 detected genes as the second filtering step.

2. It is unclear how many biological replicates of mice were used.

Response

We analyzed single-cardiomyocyte transcriptomes from 2 mice at each time point (at 2 weeks after sham operation and at 3 days and 1, 2, 4, and 8 weeks after TAC). We used echocardiography to assess whether the heart was appropriately exposed to pressure overload (**Supplementary Fig. 1a,b**). Averaged single-cell expression profiles were tightly correlated between biological replicates at each time point (**Supplementary Fig. 1j,k**). We also compared the transcriptomic profiles on the t-SNE plot, confirming the similarity between cardiomyocytes at the same time points (**Supplementary Fig. 6e**).

3. The authors performed single cell RNA-seq on Illumina HiSeq2500. However,

important details were missed out which will aid in the technical assessment

a. Paired-end or single-end

b. Read lengths

c. Insert size

Response

Single-cell libraries were subjected to paired-end 51-bp RNA sequencing on a HiSeq 2500 in rapid mode. Insert size was 345 ± 40 bp (average \pm standard deviation). These details were added to the Methods section (**page 33, line 14**).

4. Related to point 1 above, no mention of how the quality of reads was assessed. For example, the authors might have included libraries with average base quality > 30 in the downstream analysis because low base quality results in technical noise and drawing unreliable conclusions. According to the methods section, FPKM 10 was used as a threshold to call expressed genes. But in the results section, RPKM >0.1 was used. This is conflicting information. Even for the use of both FPKM and RPKM: although technical equivalently, it would be good to be consistent. Perhaps to clarify, it would be useful to be clear how FPKM or RPKM were derived. RPKM >0.1 seems low stringency, and we expect to result in significant noise: how do the authors justify this choice?

Response

We apologize for not mentioning the assessment of sequencing quality. We summarized the mean quality score of all samples (**Supplementary Tables 1, 2**). We confirmed that all samples that were passed through the filtering steps had a mean quality score > 30 .

We also apologize for the misleading expressions. We used two thresholds for selecting the cells and genes for downstream analysis; RPKM > 0.1 was used for selecting the cells (criterion for detecting genes) and RPKM > 10 was used for selecting the genes (criterion for quantitatively analyzing genes). We added this description to the Methods section (**page 34, line 1**). We do not use “FPKM” in this study.

5. It was also unclear if the authors looked at protein coding genes or non-coding genes or both. Did they derived information from UCSC, Ensembl or GENCODE?

Response

We downloaded the RefSeq transcripts (coding and non-coding) from the UCSC genome browser (<http://genome.ucsc.edu>), mapped the RNA-seq reads to the mm9 and hg19 genomes, calculated RPKM values for all RefSeq transcripts, and used them for gene expression analysis.

6. We prefer commonly used mapping software such as STAR or Tophat for alignment of RNA-seq reads. Many reads are not mapped by Bowtie1, particularly when reads span 2 exons involving splicing. Did the authors explore options?

Response

We mapped the RNA-seq reads using Tophat and Bowtie1 and compared the calculated RPKM values of sham cardiomyocytes, confirming that the averaged single-cell profiles were tightly correlated between them (**Supplementary Fig. 1h**). We also conducted WGCNA to confirm that the essential gene modules (M1, M2, M3, M5, M11, M12, M16, M22, M24, and M7) identified in this study were preserved even in single-cardiomyocyte transcriptomes obtained by using Tophat (**Supplementary Figure 6d**).

7. Major stats and QC reflecting quality of the sequencing runs are not provided. Without these stats, it is not possible to tell if the results are due to technical artifact. Some important metrics are:

- a. Library sizes*
- b. Total mapped reads*
- c. Mapping efficiency (% uniquely mapped reads)*
- d. % duplication*
- e. % reads aligned to ribosomal RNA*
- f. % reads aligned to mitochondrial DNA*
- g. % reads aligned to introns and intergenic regions*

Response

We apologize for not providing the data regarding the major statistics and sequencing quality control. We summarized these data in **Supplementary Tables 1, 2**.

8. On page 5 the authors state that the cells were isolated at different time points and, as a result, it is possible that their transcriptome is influenced by technical factors (batch effects). How did the authors test and, if needed, controlled for these effects? Batch effect is the singular most difficult technical hurdle such biologically targeted projects face. The authors will need to rigorously demonstrate how they have accounted for this.

Response

To assess the possibility of batch effects, we added ArrayControl RNA Spikes 1, 4, and 7 (Ambion, cat. No. Am1780) to the lysis buffer at the pre-defined concentrations on two separate plates and conducted single-cardiomyocyte RNA-seq of wild-type mice, confirming a good correlation between RNA spike-in concentrations and their expected RPKM values (normalized by unique reads mapped to the nuclear genome) in both batches (**Supplementary Fig. 1c**). We also performed t-SNE analysis of single-cardiomyocyte transcriptomes from normal C57BL/6 mice (RPKM values) in 2 different batches to confirm that cardiomyocytes could not be classified by batch (**Supplementary Fig. 1d**). On the basis of these findings, we considered that we did not need to control for batch effects and used RPKM normalization for quantitative gene expression analysis in this study. We added this description in the revised manuscript (**page 34, line 3**).

9. In single-cell studies it is often suggested the use of spike-in controls for data normalization. This could be especially useful for normalizing for batch effects. Did the authors include such controls and if not why?

Response

In addition to the findings from the preliminary experiments as mentioned above, we also found that there were some wells with low efficiency of reverse transcription and amplification of endogenous RNA in spite of the high efficiency of those with RNA spikes, suggesting that the use of spike-in controls for data normalization might distort endogenous RNA expression levels in our experiments. Therefore, we did not use RNA spikes for data normalization and used RPKM normalization for quantitative gene

expression analysis in this study.

10. It is not clear from the analysis of page 5 how many samples were excluded at each time point.

Response

We indicated the number of samples used during the filtering process (**Supplementary Fig. 1e**).

11. In the results section, the authors mention “expressed profiles were tightly correlated between the average expression of single cells and corresponding bulk expression” It was unclear if the authors averaged the single cell profiles across all timepoints? Were there any point estimates for the correlation? Is “between biological replicates” referring to biological replicates of bulk RNA-seq or single cell RNA-seq.

Response

We apologize for using a misleading description. Regarding the former point, we averaged the single-cell profiles at each time point and compared them with the corresponding bulk profiles (**Supplementary Fig. 1j,k**). Regarding the latter point, we compared averaged single-cell profiles of biological replicates at each time point (**Supplementary Fig. 1l,m**). We have revised the manuscript accordingly (**page 5, line 17**).

12. It is a shame that the authors did not attempt to provide a global overview of the sham-TAC single cell transcriptomic landscape before they dived into detailed network analysis. This global analysis would have allowed the analysis for:

- a. How heterogeneous are sham and TAC cells. Do they cluster well?*
- b. Pointing out obvious outliers, and / or cell type contaminations (eg fibroblasts, endothelial cells)*
- c. Correlation with potential confounding factors such as batch effect, number of expressed genes etc.*

Response

Hierarchical clustering using all genes expressed at RPKM ≥ 5 in at least 1% of all cells revealed that cardiomyocytes from sham and TAC mice were well clustered (**Supplementary Fig. 2a**). We also calculated the correlation coefficients of single-cell transcriptomes among cells at each time point and found an increase of transcriptional heterogeneity during heart failure (**Fig. 1g**). We added these descriptions in the revised manuscript (**page 5, line 23**).

We did not find any obvious outliers, but found that some genes thought to be specifically expressed in endothelial cells (*e.g.*, *Cav1* and *Pecam1*) or fibroblasts (*e.g.*, *Dcn* and *Lum*) were clustered together with genes essential for transcription; *Cav1* and *Pecam1* were in G7 and *Dcn* and *Lum* were in G8 (**Supplementary Fig. 2a**). We used single-molecule RNA *in situ* hybridization to validate that these genes are expressed in sham and TAC cardiomyocytes (**Supplementary Fig. 4**). A previous study of single-nucleus RNA-seq analysis of cardiomyocytes also mentioned the presence of cardiomyocytes expressing endothelial marker genes (See et al. *Nat Commun.* 2017), consistent with our findings. Therefore, we did not eliminate cardiomyocytes expressing these genes from the downstream analysis. We added these descriptions in the revised manuscript (**page 6, line 11**).

We performed principal component analysis to show that cardiomyocytes from different batches were located closely at every time point (**Supplementary Fig. 2b**). We also found that PC1 values were correlated with the number of expressed genes and that cells with high PC1 values were enriched for cardiomyocytes from mice at 3 days and 1 week after TAC operation. We consider that an increase in the number of detected genes at the early stage after pressure overload might reflect the expression of stress response genes and an enlargement of cell size after pressure overload. We added these descriptions in the revised manuscript (**page 5, line 24**).

13. We found the combination of random forests and hierarchical clustering tools of page 6 very interesting. However, there was a lot of information condensed in a single paragraph that makes it difficult for the reader to understand, and thus making it confusing. Expression profiles were used in WGCNA to obtain 55 modules whose PC1 were extracted. The random forests were applied to the PC1 values to select the 9 most important modules. Was the subsequent cell clustering based on the PC1 data of these 9 modules or the gene expression profiles of the genes included in these 9 modules? It is

not clear to us what would be the most accurate way to cluster the cells and we think that this depends on the variance explained by the PC1. We would have liked to see a comparison of these two alternatives and the reasoning behind the authors' preferred choice.

Response

After obtaining 9 modules by the combination of random forests and hierarchical clustering, we performed the subsequent cell clustering analysis based on the PC1 values of the 9 modules to classify cardiomyocytes. This is because the combination of random forests and hierarchical clustering showed that the accuracy of classification using the PC1 values of the 9 modules was higher than that using the gene expression profiles of the 9 modules (**Supplementary Fig. 7a-c**). A comparison of the hierarchical clustering results using the gene expression profiles with that those from using the PC1 values showed that most of the cells classified into the same cell clusters were well clustered (**Supplementary Fig. 7a**). We added these descriptions in the revised manuscript (**page 35, line 10**).

14. From supplementary figure 2b it is evident that the 9 modules give 7 random forest based clusters with low error rates. What parameters / algorithm were used to derive these 7 clusters? The random forest is supervised (based on these 7 clusters). It would be interesting to see how the error rate changes for 5-10 clusters and what clusters an unsupervised algorithm would obtain (e.g. clustering / trajectory estimation and states by Monocle).

Response

We compared the error rates for 4–10 clusters and found that they were maintained at a low level for 4–7 clusters, while a drastic increase of the error rate was observed for 8 clusters (**Supplementary Fig. 7b**). Therefore, we chose 7 clusters for cell classification. We also used Monocle to derive the trajectory information and found a relationship between cell-state identified by Monocle and cell-cluster identified by the combination of random forests and hierarchical clustering (**Fig. 2a,b**).

15. Pages 6-7 discuss the cell trajectory on the tSNE plot but we could not find any

details on how this trajectory was estimated. Results based only on visual inspection can be highly misleading due to the data noise, the optimization process and the huge amount of information included in a typical single-cell dataset. Other than that, the analysis of pages 7-8 with the module variability is very interesting but it needs to be supported by a more rigorous statistical analysis.

Response

We are grateful to the reviewer for this insightful comment. We used Monocle to derive the cardiomyocyte trajectory after TAC operation, revealed the cell-state transition during the trajectory, and identified the branch point of failing cardiomyocytes (**Fig. 2a**). Pseudo-time analysis with differential expression analysis uncovered the expression dynamics of the genes involved in the induction of failing cardiomyocytes; high expression of M3 and M24 genes and low expression of M1 and M2 genes were characteristic for failing cardiomyocytes (**Fig. 2c,d**). Along the lines of these findings, we revised the manuscript (**page 9, line 3**).

16. It is not clear how ImageJ measured the cell area. Were the estimates derived from 2D images? How was the volume of the cell assessed? What does “small” and “large” cell means in figure 2b. What is the variability in cell size and how is it correlated with the cell library size? Would spike-in or endogenous controls be a better way to normalize this dataset?

Response

We measured cell area from 2D images. Single cardiomyocyte images were obtained on dishes using an XZ-2 microscope (Olympus). We measured cardiomyocyte area in the buffer after isolation using NIH ImageJ software. The definition of a “small” and “large” cell is based on cell area in **Fig. 3d (Fig. 2d** in the original version). The data regarding the distribution of cell area are shown in **Supplementary Fig. 11e**. We also found a positive correlation between cell area and library size (the number of unique reads mapped to the nuclear genome) ($R = 0.36$, Pearson’s correlation coefficient). However, after calculating RPKM values normalized by the number of unique reads mapped to the nuclear genome, we found that mitochondrial gene expression was correlated with cell area. Even for the RPKM values normalized by the endogenous control (*Tnnt2* gene

expression), mitochondrial genes were significantly enriched in the top 300 genes correlated with cell area (**Supplementary Fig. 11f**). We selected *Tnnt2* as an endogenous control gene because real-time quantitative PCR of single-cell cDNA libraries showed a relatively homogenous expression pattern in cardiomyocyte libraries with efficient reverse transcription and amplification (**Supplementary Fig. 1f**). Therefore, we conclude that the expression of genes involved in mitochondrial ribosome and metabolism is correlated with cell area in the hypertrophy stage after pressure overload.

17. How do the authors ensure that cell sizes were maintained following harvest? Our experience suggests that cardiomyocytes are very sensitive to hypercontraction depending on calcium concentrations when isolated out ex vivo. Is there a way to validate this?

Response

As the reviewer pointed out, isolated cardiomyocytes are very sensitive to hypercontraction (**Supplementary Fig. 11a**). To prevent hypercontraction, after isolation, we resuspended the cardiomyocytes in medium (NaCl 130 mM, KCl 5.4 mM, MgCl₂ 0.5 mM, NaH₂PO₄ 0.33 mM, D-glucose 22 mM, HEPES 25 mM, fetal bovine serum 0.2%, pH 7.4) containing a low concentration of calcium (0.1 mM). To analyze the potential morphological changes of cardiomyocytes, we performed time-lapse imaging analysis of cardiomyocytes at 0, 1, 5, 30, and 60 min after isolation and confirmed that cell sizes were maintained following harvest (**Supplementary Figure 11b-d**).

18. Page 10 summarizes the logic behind network comparison but it lacks the details of the methodology followed. How were the differences of the hypertrophy / non-hypertrophy networks been statistically assessed? The information of this paragraph, the paragraph of page 12 and the supporting figures was not enough to understand the key steps of this analysis.

Response

We apologize for not writing the details. If there are hypertrophy stage-specific

networks, network analysis using single-cell transcriptomes of cardiomyocytes except hypertrophy-stage cardiomyocytes cannot identify such network modules. This idea is based on the concept for identifying sample-specific networks (Kuijjer et al. *arXiv*. 2015; Liu et al. *Nucleic Acids Res.* 2016). Therefore, we separately performed WGCNA on all cardiomyocytes and on those except hypertrophy-stage cardiomyocytes, and assessed the significance of the overlap between gene modules using Fisher's exact test with the WGCNA "overlapTable" function (Langfelder et al. *PLoS Comput Biol.* 2011; Hilliard et al. *PLoS Comput Biol.* 2012). We added this description to the revised manuscript (**page 12, line 17**).

19. There was no differential expression analysis to assess the differences across the cardiomyocyte isolation time points, across the estimated clusters and across the cell trajectory. This information is important in order to assess the significance of the findings. It could point out cell contamination (other cell types), which was not discussed in this work.

Response

We added the data of differential expression analysis to assess the differences across the time points (**Supplementary Fig. 3a**), across the cell clusters (**Supplementary Fig. 3b**), and across cell trajectory (**Fig. 2d**). As we mentioned above, we performed single-molecule RNA *in situ* hybridization to verify the possibility of cell contamination, validating the presence of cardiomyocytes expressing endothelial and fibrotic genes in both sham and TAC hearts. A previous study of single-nucleus RNA-seq of cardiomyocytes also mentioned the presence of cardiomyocytes expressing endothelial marker genes (See et al. *Nat Commun.* 2017), consistent with our findings. Therefore, we did not eliminate cardiomyocytes expressing these genes from the downstream analysis.

20. It is not clear what value H3K27ac ChIP-seq added to the understanding of the modules involved in cardiomyocyte hypertrophy.

Response

Through the integrative analysis of single-cell transcriptomes and morphology, we

identified module M1 as essential for cardiac hypertrophy, but did not know the upstream regulators of this module. We then hypothesized that the activation of M1 genes is induced by the activation of DNA elements regulating M1 gene expressions. Genome-wide H3K27ac mapping using ChIP-seq detected regulatory elements around M1 genes, where ELK1 and NRF1/2 recognition motifs were enriched, suggesting that these transcription factors and related signaling pathways are essential for M1 gene activation and cardiac hypertrophy. Collectively, the combination of single-cell RNA-seq and H3K27ac ChIP-seq enables us to identify simultaneously essential gene modules and their upstream regulators.

Despite these, the methodologies appear to have worked well to produce some real biological insights, that have been validated by concordance both within their own data (multiple seq datasets, including mouse and human, and RNA-FISH/immunostaining). Importantly, this has also been consistent with published literature. However this consistency with published literature also means that a lot of their findings are not actually all that surprising and have been reported before (morphological hypertrophy correlating with increases in genes for translation and mitochondrial biogenesis, adaptive hypertrophy followed by maladaptive and heart failure, cellular heterogeneity in hypertrophy, Mef2, ERK and NRF signalling, oxidative stress -> DNA damage -> disease progression). Admittedly, the temporal and single cell resolution of this study offers some advantages, and the datasets will provide a useful resource to researchers, assuming they are made publically available. Still, the authors did not demonstrated or explained how exactly “these results provide a potential application for assessing and regulating cardiomyocyte remodelling in cardiac hypertrophy and failure.”

Perhaps this was why the authors pushed the p53 story, the novelty of which is that p53 activation is critically important specifically at the point of transition from an adaptive hypertrophied cardiomyocyte, to a failing one. However this has two key problems.

The first problem is that a PNAS paper came out earlier this year showing that CM-specific KO of p53 makes hearts resistant to TAC (Mak et al 2017: <http://www.pnas.org/content/114/9/2331.full>). Although analysis was not CM-specific, this paper used microarray profiling to show many of the same effects (e.g. p21, Mef2, mitochondrial biogenesis and bioenergetics, glucose and fatty acid

metabolism). The authors of the current study have not cited and seem unaware of this paper. Unfortunately it detracts significantly from the novelty.

Response

Thank you for notifying a recent literature. We missed the citation of this important paper. We have now cited this study in the revised manuscript (**page 14, line 1**).

The second problem is that, with the massive amount of data from single cell analysis, the authors have not present enough evidence to justify the conclusion that transient p53 activation specifically drives the CM transition from compensated to failing. The above paper shows that p53 has physiological activity in mouse CM at all stages of health/disease. Cardioprotective transcriptional changes are present in p53 KO hearts by day 7 TAC – that is before the appearance of failing CM. It is therefore conceivable that p53 KO hearts maintain a generally higher level of health in TAC than wild type hearts do, and it is because of this that CMs in KO hearts never reach the stage of transition to failure, rather than because of a specific lack of p53 activation at that point. Perhaps if left longer than 8 weeks, failing cells would appear in p53 KO hearts.

Aside from the KO model, the authors base their evidence for p53 activation driving the failure transition on a very small number of cells. p53 activation is not specifically measured, only upregulation. It is possible that in these cells, p53 activation/upregulation is an effect of an already damaged and failing cell, rather than the cause of it. The authors make no mention of p53 upregulation in human DCM cells. Was it not detected? If not, how is this discrepancy explained?

Response

As the reviewer pointed out, it is difficult to distinguish physiological and pathological functions using a knockout model. However, there are five reasons why we insist that p53 activation underlies the induction of failing cardiomyocytes in this paper. First, p53 activation was observed at the branch point for failing cardiomyocytes during the trajectory identified using Monocle. Second, cardiomyocytes from p53CKO mice were remodeled through the branch point, but escaped the induction of failing cardiomyocytes after TAC operation (**Figs. 2a,b and 5h**). Third, recent live cell imaging experiments revealed that p53 activation is transient (Loffreda et al. *Nat Commun.*

2017), and this transient activation is sufficient for cell fate conversion (Johmura et al. *Mol Cell* 2014; Krenning et al. *Mol Cell* 2014). Fourth, our additional experiments showed that p53 is essential for Nrf2 activation in the heart failure stage and that Nrf2 directly regulates Mef2a expression (**Fig. 6e,f** and **Supplementary Fig. 14**), and provided evidence for the involvement of the p53-Nrf2-Mef2a axis in the induction of heart failure. Lastly, patients with cardiomyocytes expressing human M1 genes, which include CDKN1A, a p53 target gene, did not show a response to LVAD treatment (**Fig. 7g-j**). Collectively, although we cannot exclude the possibility that the physiological activity of p53 might affect cardiac function, considering the timing of p53 transient activation, the dynamics of state transition of p53CKO cardiomyocytes, the direct link between p53 and heart failure-related transcription factors, and the relationship with human pathogenesis, we emphasize the significance of p53 activation for the induction of heart failure.

Reviewer #2 (Remarks to the Author):

The authors performed exhaustive single-cell RNA-seq to define the changes in RNA expression that occur in response to pressure overload (TAC) of the heart in mice. A time course of TAC was performed, ranging from early stage compensatory hypertrophy (day 3 and week 1) to late stage heart failure (week 8). Distinct gene regulatory modules were defined computationally at each stage of the response. The transition from hypertrophy to failure was marked by induction of p53-dependent gene expression (Module 24 genes). Interestingly, p53 appeared to regulate Module 24 gene expression, at least in part, by stimulating MEF2 and NRF2 transcription factor-dependent gene expression. To follow up on this, the authors generated cardiomyocyte-specific p53 KO mice. These animals appeared to be protected from heart failure induced by TAC. Finally, the authors perform single-cell transcriptomics on human cardiomyocytes from normal controls and hearts from individuals with dilated cardiomyopathy.

The authors are to be commended for performing a tour de force evaluation of the transcriptional networks that are associated with cardiac hypertrophy. The datasets will provide important resources for the field. The segment of the manuscript that is

dedicated to p53 is nice because it provides a mechanistic evaluation that deviates from the somewhat encyclopedic nature of the other figures. I suggest that the authors expand on the p53-related findings to enhance the overall manuscript.

Specific points

1. For Fig. 1B, it would be useful to provide length-to-width ratios for myocytes from other time points following TAC.

Response

We showed the data for the transition of length-to-width ratios of cardiomyocytes following TAC operation (**Fig. 6e**).

2. For sFig. 5F, the authors should perform single-cell qPCR for Trp53 from WT mice post- 2 weeks TAC to rule out effects of the floxed allele on p53 expression.

Response

We are grateful to the reviewer for this important comment. We conducted single-cell quantitative PCR of cardiomyocytes from wild-type mice at 2 weeks after TAC operation, and showed an expression pattern similar to that from p53^{flx/flx} mice, ruling out possible effects of the floxed allele on p53 expression (**Supplementary Fig. 13f**).

3. The authors should quantify myocyte cross-sectional area in p53 KO hearts vs. controls, before and after TAC.

Response

We measured cardiomyocyte cross-sectional area in p53^{flx/flx} and p53CKO hearts after sham and TAC operation (**Supplementary Fig. 15**), and obtained results that were consistent with those from echocardiographic (**Fig. 5b**) and single-cardiomyocyte morphological (**Fig. 6h**) assessments.

4. The manuscript would be enhanced by experiments that address the mechanisms by which p53 regulates MEF2 in cardiomyocytes. To my knowledge, such findings would

be highly novel.

Response

As mentioned later, we demonstrated that Nrf2 protein level in the heart was increased in the heart failure stage after TAC operation, which was blocked in p53CKO mice (**Fig. 6e** and **Supplementary Fig. 14a-d**). This finding suggests that p53 mediates Nrf2 protein activation during heart failure. Since public Nrf2 ChIP-seq data of macrophages showed that Nrf2 binds to the Mef2a promoter region, which contains an Nrf2 recognition motif (Eichenfield et al. *eLife* 2016) (**Supplementary Fig. 14e**), we hypothesized that activated Nrf2 directly regulates Mef2 gene expression in cardiomyocytes. We conducted ChIP-qPCR of TAC cardiomyocytes to show a significant enrichment of Nrf2 at the Mef2a promoter (**Fig. 6f**), validating our hypothesis that the p53-Nrf2-Mef2a axis is essential for the induction of failing cardiomyocytes. We revised the manuscript (**page 16, line 2**).

5. The authors mention prior work showing p53-induced NRF2 activation via p21-mediated NRF2 protection against Keap1-mediated ubiquitination. They should determine if this mechanism is blocked in p53 KO hearts.

Response

We performed western blot analysis of heart tissues from p53^{flox/flox} and p53CKO mice after sham operation and at 8 weeks after TAC operation (heart failure stage), and validated that Nrf2 protein level was increased after TAC operation in p53^{flox/flox} mice, which was blocked in p53CKO mice (**Fig. 6e** and **Supplementary Fig. 14a-d**). We are very grateful to the reviewer for this valuable comment. We revised the manuscript (**page 16, line 2**).

Reviewer #3 (Remarks to the Author):

This is a valuable manuscript studying how cardiomyocytes undergo molecular and morphological changes in response to stress, leading to cardiac hypertrophy and failure. The authors relate these changes to gene expression profiles measured using single-cell

mRNA-Seq at cardiomyocytes samples collected at different time points representing disease progression.

The authors find interesting relationships between gene modules and trajectories of disease progression. Gene modules represent sets of genes that show similar expression and are significantly expressed across cells. The identified modules include important genes known from literature to be responsible for the cardiac hypertrophy and the heart failure.

Overall, the structure and sequence of the results is well presented and written. It starts by investigating the responsible genes for early stage hypertrophy, then for the progression to heart failure. Based on their data, they conclude p53 being responsible for the heart failure progression, which they validate using a p53 blocking mouse. Finally, they validate these responsible gene modules in human tissues.

The analysis of the single-cell mRNA-Seq data is well structured and is mainly based on the WGCNA R package pipeline that is widely known and used in gene co-expression analysis. The authors combined these analyses with various visualization tools to produce clear figures, as well as function enrichment analysis to interpret the observed correlations between the observed important genes and the disease progression.

In conclusion, I am very happy with this manuscript. I do have a few comments that I would like to see addressed.

Major comments:

1. In the hierarchical clustering of co-expressing genes into gene modules for the data of the C57BL/6 mouse model, the dendrogram in Figure 1c does not support the 7 cluster assignments mentioned. Further clarification is needed if there are any other criteria for clusters assignment.

Response

To achieve accurate cell-type classification, in **Fig. 1d** (**Fig. 1c** in the original version), we used hierarchical clustering (minimum cluster size of 5% of all samples) and random

forests to compare the error rates for 4–10 clusters, and found that they were maintained at a low level for 4–7 clusters, while a drastic increase of the error rate was observed for 8 clusters (**Supplementary Fig. 7b**). Therefore, we chose 7 clusters for cell classification. We added this description to the manuscript (**page 35, line 10**).

2. In the analysis of the hypertrophy related modules, Figure 2b shows a scatter plot of PC1 and PC2 with only 34 cells (dots) while the authors previously mentioned that in week one 82 cardiomyocytes are obtained. Why is there a difference?

Response

We apologize for misleading the reviewer. In **Fig. 3b (Fig. 2b** in the original version of the manuscript), we again isolated cardiomyocytes at 1 week after TAC operation to assess their morphology and transcriptome simultaneously. We revised the manuscript accordingly (**page 10, line 18**).

Minor comments:

1. For the purpose of reproducibility of the analysis, please specify the perplexity used for producing the t-SNE maps, as well as the distance measure and linkage method used for the hierarchical clustering.

Response

We used t-SNE with a perplexity of 10 and hierarchical clustering with correlation distance and complete linkage. We added this description to the manuscript (**page 35, line 24**).

2. Page 34, line 5, remove “at 2 weeks”. It is already mentioned in line 6.

Response

We revised the manuscript accordingly (**page 32, line 10**).

Conclusion:

I recommend this work to be published after addressing my comments.

Finally, the authors would like to thank the reviewers again for these valuable comments and suggestions.

Reviewers' comments:

Reviewer #1 (Remarks to the Author):

The authors have made some significant edits and revision of analysis to their previous submission. It is a manuscript that is very dense now with multiple angles of analysis. So much that can easily generate confusion and disorganisation of thought. I don't have much problem with the biological follow-up relating to the latter part of their analysis. But their dataset QC remains worrying because several key aspects point to an inadequate quality of the sequenced material, and therefore the subsequent bioinformatics analysis that has been presented.

As major examples:

1. Line 97: ≥ 5000 genes were detected at the low RPKM >0.1 . RPKM > 0.1 is a low and lenient threshold, and to discover only ~ 5000 expressed genes suggests that the dataset is not robust. Based on this number, there would be far fewer expressed genes expected at the more usual threshold of RPKM >1 ! This could really mean that the data quality of this single-cell RNA-seq set is bad, and all the downstream analysis are not convincing to interpret.
2. Supplementary Figure 1h. Why did the authors persist to use their Bowtie1 mapping results for analysis, when the authors had also done mapping and analysis using Tophat2. See the scatterplot (pointed out in figure attached), despite having high correlation, there are too many points lying on the top left of unity. This implies that there are too many genes found to be lowly expressed with Bowtie1, but yet found to be highly expressed in Tophat2. Moreover the axis is in log-scale! It is worrying that at least 100 genes were estimated to be 10-fold lower than they are in the comparison of both mapping tools. What are these genes. Was subsequent analysis built on this under-estimated values? Was the low number of genes detected (point 1 above) due to the puzzling choice of Mapping algorithm.

3. Line 127: Cell-to-cell heterogeneity increased after pressure overload. This seems to be an afterthought and the idea was not further developed. How did the authors computed the "heterogeneity"?
4. Supplementary Figure 5h. This is another key worrying point. Why are so many highly expressed genes possessing such high coefficient of variation (figure here also). This is unusual. Eg why does "Nppa" have such high coefficient of variation. Such levels of variations are more usual to see for lowly expressed genes. It raises the question of whether this is a computation error, or again the robustness of the quality of the sequenced material. For *Nppa*, this means that some TAC cells aren't expressing *Nppa*, while some TAC cells are expressing extremely high level of *Nppa*. What is the fraction of cells not expressing *Nppa* in TAC? Are these cells cardiomyocytes? Also, the authors is only showing this graph for 1 time point (Week 8). The authors should show the graph for all Time points.

Overall, there remain technical issues which cannot be adequately explained in the dataset.

Reviewer #2 (Remarks to the Author):

The authors have sufficiently addressed my original comments.

Reviewer #3 (Remarks to the Author):

The authors have addressed all the comments with convincing answers. I recommend this work to be published.

Responses to the reviewers' comments

We thank the reviewers for their insightful comments regarding our manuscript. Along the lines suggested by the reviewers, we have performed additional analyses and revised the manuscript. The sentences revised according to the reviewers' comments are highlighted in red in the main text.

Response to Reviewer #1

The authors have made some significant edits and revision of analysis to their previous submission. It is a manuscript that is very dense now with multiple angles of analysis. So much that can easily generate confusion and disorganisation of thought. I don't have much problem with the biological follow-up relating to the latter part of their analysis. But their dataset QC remains worrying because several key aspects point to an inadequate quality of the sequenced material, and therefore the subsequent bioinformatics analysis that has been presented.

As major examples:

1. Line 97: ≥ 5000 genes were detected at the low $RPKM > 0.1$. $RPKM > 0.1$ is a low and lenient threshold, and to discover only ~ 5000 expressed genes suggests that the dataset is not robust. Based on this number, there would be far fewer expressed genes expected at the more usual threshold of $RPKM > 1$! This could really mean that the data quality of this single cell RNA-seq set is bad, and all the downstream analysis are not convincing to interpret.

Response

We detected more than 5,000 genes with $RPKM > 0.1$ in 82.1% of cardiomyocytes (396/482 cells) and more than 4,000 genes with $RPKM > 1$ in 86.9% of cardiomyocytes (419/482 cells) (**Supplementary Fig. 1j**). Expression levels were comparable between cardiomyocytes that had more than 5,000 genes with $RPKM > 0.1$ and those that had more than 4,000 genes with $RPKM > 1$ (**Supplementary Fig. 1k**). Among all of the cardiomyocytes that had more than 5,000 genes with $RPKM > 0.1$, over 4,000 genes were detected with $RPKM > 1$ (**Supplementary Fig. 1l,m**). Many single-cell RNA-seq studies with the Smart-seq2 protocol have used a threshold of $RPKM > 0.1$ for gene detection

(Xue Z et al. Genetic programs in human and mouse early embryos revealed by single-cell RNA sequencing. *Nature*; Hu Y et al. Simultaneous profiling of transcriptome and DNA methylome from a single cell. *Genome Biol.* 2016; Nichterwitz S et al. Laser capture microscopy coupled with Smart-seq2 for precise spatial transcriptomic profiling. *Nat Commun.* 2016; Gao Y et al. Protein Expression Landscape of Mouse Embryos during Pre-implantation Development. *Cell Rep.* 2017). Therefore, we set a more stringent threshold of RPKM > 0.1 with 5,000 detected genes for selecting the cells in this study. We also used a threshold of RPKM > 10 to select genes for quantitative downstream analysis. We added this description to the revised manuscript (**page 23, line 22**).

2. Supplementary Figure 1h. Why did the authors persist to use their Bowtie1 mapping results for analysis, when the authors had also done mapping and analysis using Tophat2. See the scatterplot (pointed out in figure attached), despite having high correlation, there are too many points lying on the top left of unity. This implies that there are too many genes found to be lowly expressed with Bowtie1, but yet found to be highly expressed in Tophat2. Moreover the axis is in log-scale! It is worrying that at least 100 genes were estimated to be 10-fold lower than they are in the comparison of both mapping tools. What are these genes. Was subsequent analysis built on this under-estimated values? Was the low number of genes detected (point 1 above) due to the puzzling choice of Mapping algorithm.

Response

By comparing RPKM values obtained using Tophat2 and Bowtie1, we identified only 25 genes that had 10-fold higher Tophat2 mapping RPKM values (**Supplementary Fig. 1h,i**). Gene ontology analysis revealed that ribosomal protein genes were enriched in these genes ($p = 7.9e-6$). It has been reported that alternative splicing of ribosomal protein mRNAs is a conserved regulatory mechanism for maintaining translational homeostasis (Mitrovich et al. Unproductively spliced ribosomal protein mRNAs are natural targets of mRNA surveillance in *C. elegans*. *Genes Dev.* 2000; Takei et al. Evolutionarily conserved autoregulation of alternative pre-mRNA splicing by ribosomal protein L10a. *Nucleic Acids Res.* 2016) and is regulated by introns (Parenteau et al. Introns regulate the

production of ribosomal proteins by modulating splicing of duplicated ribosomal protein genes. *Cell*. 2011). This is because Tophat2 can align reads related to alternative splicing events, whereas Bowtie1 cannot. However, by comparing the numbers of detected genes mapped using Tophat2 or Bowtie1 for each cell, we found that using the RPKM values mapped by Tophat2 did not increase the number of detected genes (**Supplementary Fig. 1n**). Furthermore, in the recent literature, many researchers, including Aviv Regev and Jay Shendure, use Bowtie1 mapping RPKM values for single-cell gene expression analysis (Raj et al. Simultaneous single-cell profiling of lineages and cell types in the vertebrate brain. *Nat Biotechnol*. 2018; Harber et al. A single-cell survey of the small intestinal epithelium. *Nature*. 2017; Singer et al. A distinct gene module for dysfunction uncoupled from activation in tumor-infiltrating T cells. *Cell*. 2017). We also conducted WGCNA to confirm that the essential gene modules (M1, M2, M3, M5, M11, M12, M16, M22, M24, and M7) identified in this study were preserved even in single-cardiomyocyte transcriptomes obtained by using Tophat2 mapping (**Supplementary Fig. 6a**). Therefore, we think that the use of Bowtie1 mapping RPKM values for single-cell gene expression analysis in this study is suitable.

3. Line 127: Cell-to-cell heterogeneity increased after pressure overload. This seems to be an afterthought and the idea was not further developed. How did the authors compute the “heterogeneity”?

Response

We assessed transcriptional heterogeneity among cells by using Pearson’s correlation coefficient, which was calculated for each time point using all genes expressed with $RPKM \geq 10$ in at least 20% of the samples (**Fig. 1c**) (**page 24, line 21** in the Methods section). The increased cell-to-cell heterogeneity after pressure overload is considered to be related to bifurcation into distinct cell fates (State 2 and State 3 in **Fig. 2a**). This description was added to the revised manuscript (**page 9, line 8**).

4. Supplementary Figure 5h. This is another key worrying point. Why are so many highly expressed genes possessing such high coefficient of variation (figure here also). This is

unusual. Eg why does “Nppa” have such high coefficient of variation. Such levels of variations are more usual to see for lowly expressed genes. It raises the question of whether this is a computation error, or again the robustness of the quality of the sequenced material. For Nppa, this means that some TAC cells aren’t expressing Nppa, while some TAC cells are expressing extremely high level of Nppa. What is the fraction of cells not expressing Nppa in TAC? Are these cells cardiomyocytes? Also, the authors are only showing this graph for 1 time point (Week 8). The authors should show the graph for all Time points. Overall, there remain technical issues which cannot be adequately explained in the dataset.

Response

Actually, the expression level of *Nppa* was highly heterogeneous compared with that of *Tnnt2* in cardiomyocytes after pressure overload (**Supplementary Fig. 5b**). All cells expressed *Tnnt2* homogeneously, indicating that all of these cells were cardiomyocytes. After pressure overload, some cardiomyocytes expressed *Nppa* mRNA at a high level, whereas some cells showed low *Nppa* mRNA expression. We also used smFISH to quantify the single-cell mRNA levels of *Myh7* (high variability) and *Atp2a2* (low variability), validating the cell-to-cell variation in gene expression obtained from single-cell RNA-seq (**Supplementary Fig. 5c-e** and **Supplementary Table 3**). We also included graphs demonstrating the variability of gene expression in cardiomyocytes for all time points (**Supplementary Fig. 5a**).

Reviewer #2 (Remarks to the Author):

The authors have sufficiently addressed my original comments.

Reviewer #3 (Remarks to the Author):

The authors have addressed all the comments with convincing answers. I recommend this work to be published.

Finally, the authors would like to thank the reviewers again for these valuable comments and suggestions.

REVIEWERS' COMMENTS:

Reviewer #1 (Remarks to the Author):

The authors have addressed my concerns. I am ready to accept this publication.

Responses to the reviewers' comments

We thank the reviewers for their insightful comments regarding our manuscript.

Response to Reviewer #1

Reviewer #1 (Remarks to the Author):

The authors have addressed my concerns. I am ready to accept this publication.

Response:

The authors would like to thank the reviewers again for these valuable comments and suggestions